# UniFLoW: Universal Multi-Modal Federated LoRA Fine-Tuning Framework with Analytical Aggregation

**Haoyuan Liang** [1,2]  **Zhiyu Ye** [3]  **Jielong Tang** [1]  **Yang Yang** [1]  **Shilei Cao** [1,2]  **Guowen Li** [1,2]  **Fei Hu** [4]
**Zhiwei Zhang** [1,2]  **Haohuan Fu** [3]  **Juepeng Zheng** [1,2]

## Abstract

As Multimodal Large Language Models (MLLMs) continue to be trained, the availability of public data diminishes, limiting the possibility for further training and adaptation. However, private data remains an underutilized yet valuable resource. Federated Learning (FL) enables decentralized training on private data, yet extending it to MLLMs is challenging: heterogeneous client modalities induce architectural incompatibility, and full-parameter fine-tuning of billion-scale models incurs prohibitive communication costs. Parameter-efficient methods like LoRA alleviate these issues but introduce aggregation inconsistency, as averaged low-rank updates fail to recover the true global update faithfully. To address these issues, we propose ❄UniFLoW (**Uni**versal multi-modal **F**ederated **Lo**RA fine-tuning framework **W**ith Analytical Aggregation), a unified federated framework that leverages pre-trained large language models and multi-modal Encoder architecture, and our proposed **Fed**erated **A**ggregating **A**nalytical **Lo**w-**R**ank **A**daption ($FedA^2$-LoRA). **Uni-FLoW** effectively utilizes fragmented client-side multi-modal data while $FedA^2$-LoRA ensuring consistent aggregation. And modality-specific encoders and a II stage training strategy ensure effective integration of diverse modalities without overfitting. Experiments on text, image, and speech demonstrate that **UniFLoW** enables scalable, communication-efficient, and aggregation-consistent federated fine-tuning, with $FedA^2$-LoRA achieving state-of-the-art

performance compared to existing FedLoRA approaches. We envision **UniFLoW** as a promising solution to the growing scarcity of public data.

## 1. Introduction

Multimodal Large Language Models (MLLMs) have achieved significant advancements, primarily fueled by pre-trained models (Min et al., 2023; Li et al., 2024) on large multimodal datasets. However, with diminishing public data availability, accessing diverse private multi-modal data becomes crucial for continued model adaptation. Benefiting from the rapid development of mobile devices (Mairittha et al., 2020), personal data has become abundant, offering a vast resource for MLLMs pre-training and fine-tuning. Yet, such data often contains sensitive private information, which clients are reluctant to share. For example, a street-view image may inadvertently reveal a personal license plate number.To address these privacy concerns, Federated Learning (FL) (McMahan et al., 2017; Liang et al., 2025b) allows clients to train models locally on private data while sharing and aggregating models to fully exploit these resources. Despite its effectiveness, traditional FL restricts parameter aggregation to unimodal models. In the real world, clients often possess heterogeneous multimodal data (as illustrated in Figure 1(a)). ② *This modality inconsistency induces architectural incompatibility, thereby rendering existing parameter aggregation approaches infeasible in multimodal scenarios.*

Another prominent challenge of applying FL to MLLMs lies in ③ *communication overhead*. Given that MLLMs typically contain tens of billions of parameters, full-parameter fine-tuning (Lv et al., 2023) not only incurs substantial communication costs—since an enormous number of parameters must be transmitted to the server—but also aggravates issues such as knowledge forgetting and overfitting, as shown in Figure 1 (b). To address this issue, incorporating Parameter-Efficient Fine-Tuning (PEFT) (Han et al., 2024) into Federated Large Language Models (FedLLMs) (Ye et al., 2024a) offers a promising and scalable solution. Among these PEFT methods, LoRA-based

[1]Sun Yat-sen University, Zhuhai, China [2]National Supercomputing Center in Shenzhen, Shenzhen, China [3]Tsinghua International Graduate School, Shenzhen, China [4]South China University of Technology, Guangzhou, China. Correspondence to: Juepeng Zheng <zhengjp8@mail.sysu.edu.cn>.

*Proceedings of the $43^{rd}$ International Conference on Machine Learning*, Seoul, South Korea. PMLR 306, 2026. Copyright 2026 by the author(s).

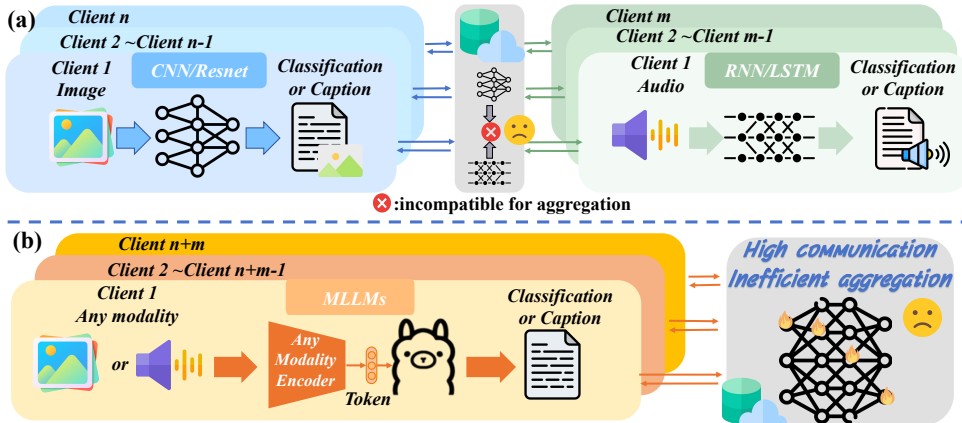

*Figure 1.* (a) Traditional FL faces architecture heterogeneity: clients handling different modalities (e.g., images with CNNs and audio with RNNs) employ incompatible model structures, making parameter aggregation infeasible. (b) A unified architecture based on MLLMs can process diverse modalities through modality encoders. However, full-parameter training and transmission introduce high communication costs and inefficient aggregation.

(Hu et al., 2022; Cho et al., 2024) methods have become increasingly popular. Although LoRA offers significant advantages in training efficiency and model generalization, it introduces aggregation inconsistency in federated settings. Specifically, when aggregating LoRA updates from multiple clients, the global weight update $\Delta_W$ computed by averaging the low-rank components and then performing matrix multiplication deviates from the true average of the local updates $\Delta_W^*$, as shown below:

$$④ \; \Delta_W = \left( \tfrac{1}{K} \sum_{k=1}^{K} \boldsymbol{B}_k \right) \left( \tfrac{1}{K} \sum_{k=1}^{K} \boldsymbol{A}_k \right)$$

$$\neq \Delta_W^* = \tfrac{1}{K} \sum_{k=1}^{K} \boldsymbol{B}_k \boldsymbol{A}_k$$

Recently, several FedLoRA (Wu et al., 2024a) variants have been proposed to address this issue. Most of them focus on uploading a single matrix, either $\boldsymbol{A}$ or $\boldsymbol{B}$, as the optimal choice. For example, one line of work uploads only one of these matrices to the server, which offers a certain level of personalization but still cannot fully recover the complete global update $\Delta_W$. Another line of work (Singhal et al., 2024) directly transmits the residual matrix $\Delta_W - \Delta_W^*$, but this significantly increases communication costs compared to exchanging low-rank updates, thereby weakening the efficiency advantage of LoRA. ⑤ *Current FedLoRA methods still fail to achieve an effective balance between recovery accuracy and communication efficiency.*

To address the challenges of ① *unlocking private multimodal data at scale*, ② **modality-induced architectural incompatibility**, ③ *communication overhead*, ④ **aggregation inconsistency**, and ⑤ *accuracy–efficiency trade-off* in exploiting private multimodal client data, we propose the **Uni**versal multi-modal **F**ederated **Lo**RA fine-tuning framework **W**ith Analytical Aggregation (**UniFLoW**). Building on the FL and using modality-specific encoders (Im-

ageBind (Girdhar et al., 2023)) together with a shared base model (Vicuna-7B (Zheng et al., 2023)) resolves ① and ②,in **UniFLoW**. This design enables the system to process up to six modalities and seamlessly scale with additional encoders, while inserting LoRA adapters into both the encoders and the base model so that only a small number of trainable parameters need to be communicated across federated rounds.

Within the **UniFLoW** framework, we further develop **Federated Aggregating Analytical Low-Rank Adaptation** ($FedA^2$**-LoRA**) as the core analytical aggregation module to address ③, ④, and ⑤ for LoRA-based FedLLMs. Inspired by (Guo et al., 2024)'s analysis of the optimal solution and our gradient analysis (in the Appendix C), we observe that the $A$ matrices tend to capture more global and stable directions than the $B$ matrices. Based on this observation, $FedA^2$-LoRA aggregates the $\boldsymbol{A}$ matrices across clients via averaging to obtain $\boldsymbol{A}^{t+1}$, and then analytically reconstructs the corresponding $\boldsymbol{B}^{t+1}$ by solving a regularized least-squares (Ridge Regression) problem (Zhang et al., 2010) so that the resulting low-rank update $\boldsymbol{B}^{t+1}\boldsymbol{A}^{t+1}$ closely approximates the desired full-rank update $\Delta_W^*$. Compared to FedExLoRA (Singhal et al., 2025), which relies on transmitting residuals, $FedA^2$-LoRA not only improves recovery accuracy but also reduces communication costs, thereby providing a more consistent and efficient federated aggregation scheme for LoRA parameters.

Furthermore, in real-world multimodal federated learning (Feng et al., 2023) scenarios, each client typically possesses only one modality, and different clients have different modalities. For instance, some hospitals may only have ultrasound data, while others may only have X-ray data. This diversity complicates modality alignment in FL for **UniFLoW**. To prevent LLMs from over-integrating

modality-specific information and neglecting content information during training, we propose an adaptive II stage training, where we first fine-tune only the modality encoders with the base model frozen, and then, after a warm-up period, switch to fine-tuning only the base model with the encoders frozen. This approach ensures a more balanced and effective learning process by gradually incorporating both modality-specific and global model updates.

In our multimodal experiments, we involved speech, images, and text. We demonstrated that **UniFLoW**'s visual modality training can enhance the LLM and facilitate audio QA tasks. Furthermore, multimodal client-side training continuously improves the LLM's understanding capabilities. UniFLoW is the first framework to fine-tune FedMLLM across three modalities. In QA tasks, $FedA^2-LoRA$ consistently restores global updates while minimizing communication costs, outperforming existing methods.

Our main contributions are summarized as follows:

- To fully harness the potential of distributed multi-modal private data, we present UniFLoW, a novel federated fine-tuning framework for Multi-modal Large Language Models (MLLMs), designed to support a wide range of modalities.

- To address the aggregation inconsistency in federated LoRA fine-tuning, we propose $FedA^2$-LoRA, which retains the parameter-efficient nature of LoRA without introducing additional communication costs. Furthermore, $FedA^2$-LoRA enables faithful reconstruction of the global update $\Delta_W$, thereby substantially mitigating the inconsistency issue.

- Through extensive experiments, we validate the effectiveness of our proposed multi-modal fine-tuning framework. In addition, our $FedA^2$-LoRA consistently achieves state-of-the-art performance in both multi-modal and single-modal settings.

## 2. RELATED WORK

### 2.1. Multimodal in federated learning

The rapid progress of multi-modal learning (Huang et al., 2021; Ye et al., 2025) has significantly broadened the application scope of artificial intelligence. Federated learning (Huang et al., 2024; Liang et al., 2026), initially explored in single-modality scenarios, has also recently begun to extend to multi-modal tasks. Existing methods(Liang et al., 2025a; Qi et al., 2025) can be grouped into two categories. The first category addresses **multiple independent single-modality tasks**. For example, QFL (Pokharel et al., 2025) performs classification across different modalities such as speech and images, yet each task remains modality-specific. The second category targets **a single multi-modal task**. For instance, FedCola (Sun et al., 2024a) combines complementary local training with collaborative global aggregation to enable cross-modal knowledge sharing, achieving strong performance on image-to-text generation. Similarly, MLLM-LLaVA-FL (Zhang et al., 2025) leverages large models to supervise and guide the training of smaller federated models, enhancing vision-to-language capabilities (Liu et al., 2024). Despite these advances, current FL approaches typically operate with relatively small models and remain confined to a single multi-modal task, limiting their ability to generalize across multiple tasks or scale seamlessly to new modalities. **To the best of our knowledge, a unified and extensible federated framework capable of supporting various multimodal tasks simultaneously has yet to be developed.**

### 2.2. LoRA in federated learning

Fine-tuning large models has proven effective in adapting them to become task-specific experts (Panigrahi et al., 2023). However, in FL, full-parameter fine-tuning (FPFT) (Bian et al., 2025) incurs prohibitive communication costs, and the limited data available on each client further increases the risk of overfitting. As a result, recent research on FedLLM (Ye et al., 2024b) has increasingly shifted toward PEFT (Chua et al., 2023; Rong et al., 2025) to overcome these challenges.

Due to LoRA's (Hu et al., 2022) low training cost and strong generalization ability, it has become the predominant fine-tuning strategy for FedLLMs. FedLoRA (Yi et al., 2023) first introduced LoRA into FL, substantially reducing both communication and computation costs while mitigating overfitting. Building on this, FDLoRA (Qi et al., 2024) employed two parallel LoRA modules—one for capturing personalized information and the other for modeling global knowledge—but this design nearly doubled the computation overhead. To further reduce costs, FFA-LoRA (Sun et al., 2024b) proposed freezing the $A$ matrix and uploading only $B$, which alleviates the inconsistency of $\Delta_W$ and $\Delta_W^*$. However, this approach decreases the number of trainable parameters and places excessive reliance on the initialization of $A$. To address this, FedSA-LoRA (Guo et al., 2024) suggested uploading only $A$ for aggregation while keeping both $A$ and $B$ trainable locally, partially alleviating the dependence on the initial $A$. While this strategy enables the training of personalized models, it remains insufficient for producing a stronger global model. Moreover, although FedSA-LoRA claims that $B$ is client-data-specific, it does not conclusively demonstrate that $B$ depends solely on local data. Therefore, a more effective LoRA aggregation strategy remains necessary.

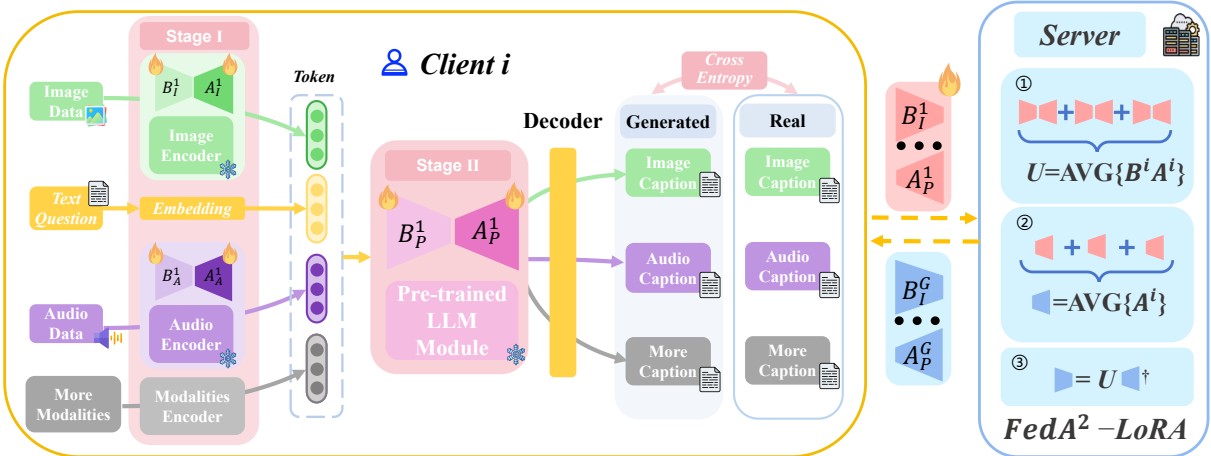

*Figure 2.* Overview of the 🌐**UniFLoW**. The inference process operates on each client, where local multi-modal data (e.g., text, images, or audio) is processed by modality-specific encoders (e.g., ImageBind), and the resulting features are then passed to a pretrained LLM (e.g., Vicuna-7B). The fine-tuning procedure is organized into two stages: in the I stage, we apply LoRA to the modality encoders, and in the II stage, we use LoRA to fine-tune the pretrained large language model. The server aggregates parameter updates from clients using the proposed $FedA^2$-LoRA method, which ensures consistent and efficient updates while minimizing communication overhead. This framework is designed to handle heterogeneous data sources across clients, thereby enhancing scalability and model performance across different modalities.

## 3. method

To exploit the fragmented and diverse modal information available on clients, we propose a Universal multi-modal Federated LoRA fine-tuning framework With Analytical Aggregation (🌐**UniFLoW**), as shown in Figure 2. We describe our **UniFLoW** from two perspectives: *client training* and *server aggregation*.

### 3.1. Training Process on the Client Side

Unlike previous FedMLLMs approaches, **UniFLoW** focuses on heterogeneous modalities, where client modalities are inconsistent. For instance, in our experiments, some clients only have audio-based Q&A or image-based Q&A, and some have both. Therefore, for client $k$, there is a dataset $D_k = \{(\mathcal{M}_{n_k}^m, Q_{n_k}, \mathcal{A}_{n_k})\}_{n=1}^N$, where $m \in M_{avail} = \{\text{image}, \text{audio}, \dots\}$ represents the modality.

For the encoder (Imagebind) and the base model (Vicuna), we not only apply LoRA to the base model, but also to the modal encoder. Any layer $\ell$ satisfies:

$$\tilde{W}^{(m,\ell)} = W_0^{(m,\ell)} + \alpha \boldsymbol{B}^{(m,\ell)} \boldsymbol{A}^{(m,\ell)},$$
$$h^{(m,\ell+1)} = \phi\left(\tilde{W}^{(m,\ell)} h^{(m,\ell)} + b^{(m,\ell)}\right) \quad (1)$$

Where $W_0$ represents the frozen parameters of the base model, $\alpha$ denotes the scaling parameter indicating the degree of injection, and $\boldsymbol{B}$ and $\boldsymbol{A}$ represent the trainable parameters. $h$ is the input feature.

For a client data sample $(\mathcal{M}_{n_k}^m, Q_{n_k}, \mathcal{A}_{n_k})$, the modal data $\mathcal{M}_{n_k}^m$ will enter Imagebind $f_m$ to process and produce the

feature $Z_{n_k}^m$. Here, $\Theta_0^{(m)}$ represents the set of frozen parameters $W_0^{(m,\ell)}$, and $\Phi_{E,k}^{(m)}$ represents the set of trainable parameters $\boldsymbol{B}_{E,k}^{(m,\ell)}$ and $\boldsymbol{A}_{E,k}^{(m,\ell)}$. As shown in Eq. 2:

$$Z_{n_k}^m = f_m(\mathcal{M}_{n_k}^m; \Theta_0^{(m)}, \Phi_{E,k}^{(m)}),$$
$$\Phi_{E,k}^{(m)} \triangleq \{\boldsymbol{B}_{E,k}^{(m,\ell)}, \boldsymbol{A}_{E,k}^{(m,\ell)}\}_{\ell=1}^{L_m} \quad (2)$$

Then, the modal feature $Z_{n_k}^m$ is mapped to a unified low-dimensional space through a pooling layer, and then projected to the dimension required by the LLM via a trainable fully connected layer $\boldsymbol{P} \in \mathbb{R}^{d_{\text{LLM}} \times d_{\text{IB}}}$. The layer $\boldsymbol{P}$ learns to bridge the gap between different modal pooling outputs and the LLM, thereby producing the feature $\mathcal{X}_{n_k}^m$ required by the LLM. Specifically, as shown in Eq. 3:

$$\mathcal{X}_{n_k}^m = \boldsymbol{P}(\text{Pool}(Z_{n_k}^m)) \quad (3)$$

Before feeding into the LLMs, we combine the question $Q_{n_k}$ with the feature $\mathcal{X}_{n_k}^m$ extracted from modality $\mathcal{M}_{n_k}^m$ to construct the prompt. Specifically, multiple modal features are concatenated and then joined with the common prompt tokens $e_{\text{BOS}}, P_{\text{before}}, P_{\text{gap}}, P_{\text{after}}$ (as shown in Appendix D) to form the final prompt $X_{n_k}$, as shown in Eq. 4:

$$X_{n_k} = \left[e_{\text{BOS}}, P_{\text{before}}, \bigoplus_{m \in M_{\text{avail}}} \mathcal{X}_{n_k}^m, P_{\text{gap}}, Q_{n_k}, P_{\text{after}}\right] \quad (4)$$

The LLMs processes the prompt $X_{n_k}$ like Eq. 2. In this formulation, $\mathcal{F}(\cdot)$ denotes the Vicuna model, $v$ is used to distinguish between the ImageBind parameters and the Vicuna parameters, and $\ell$ indicates the layer where LoRA is

inserted. Here, $\Theta_0^{(v)}$ represents the set of frozen parameters, while $\Phi_{LLM}^{(v)}$ denotes the set of trainable parameters $\boldsymbol{B}$ and $\boldsymbol{A}$. As shown in Eq. 5:

$$
\begin{aligned}
H_{n_k} &= \mathcal{F}(X_{n_k}; \Theta_0^{(v)}, \Phi_{LLM,k}^{(v)}), \\
\Phi_{LLM,k}^{(v)} &\triangleq \{\boldsymbol{B}_{LLM,k}^{(v,\ell)}, \boldsymbol{A}_{LLM,k}^{(v,\ell)}\}_{\ell=1}^{L_v}
\end{aligned}
\tag{5}
$$

The hidden state $H_{n_k}$ generated from the $(\mathcal{M}_{n_k}^m, Q_{n_k}, \mathcal{A}_{n_k})$ is mapped to a probability distribution $p_t$ through the projection matrix $w_o$ and the $\mathrm{softmax}(\cdot)$ operation, representing the likelihood of each token in the vocabulary. The true token $y_t$ is provided by the answer $\mathcal{A}_{n_k} = \{y_1, y_2, \dots\}$, and the loss between the predicted probability distribution $p_t$ and the $y_t$ is computed as shown in Eq. 6:

$$
\begin{aligned}
\mathcal{L} &= -\sum_{t=1}^{T} \log p_t(y_t \mid y_{<t}, \{X_{n_k}\}_{m \in M_{\text{avail}}}), \\
p_t &= \mathrm{softmax}(w_o(H_{n_k}^t))
\end{aligned}
\tag{6}
$$

Previously, training of MLLMs was centralized, where data from different modalities were jointly shuffled and optimized in a single pipeline, naturally aligning their representations and yielding balanced performance (e.g., Image-Bind). In FL, however, each client may hold only a single or highly biased modality, so directly fine-tuning the base model on such inputs makes it overfit modality-specific patterns rather than content-level semantics, thus harming cross-modal generalization.

To address this, we adopt a *two-stage training process*, as shown in Eq.7. When the number of local iterations $step$ is less than $\tau$, we update only the encoder parameters so that they absorb modality-specific biases and adapt to the local input distribution. When $step \geq \tau$, we update only the LLM, allowing the LLM to focus on content-related modeling based on already-normalized multimodal features. This staged schedule decouples modality adaptation from content learning and mitigates the risk of the LLM collapsing to a single dominant modality in federated settings.

$$
\begin{aligned}
\Phi_k^{(t)} &= \{\Phi_{E,k}^{(t)}, \Phi_{LLM,k}^{(t)}\} \\
&= \begin{cases} \Phi_{E,k}^{(t)} = \Phi_{E,k}^{(t-1)} - \eta \cdot \nabla \Phi_{E,k}\mathcal{L} & , if\ step < \tau \\ \Phi_{LLM,k}^{(t)} = \Phi_{LLM,k}^{(t-1)} - \eta \cdot \nabla \Phi_{LLM,k}\mathcal{L}, & if\ step \geq \tau \end{cases}
\end{aligned}
\tag{7}
$$

### 3.2. Aggregation Process on the Server Side

In FL, as shown in Eq.8, direct aggregation can introduce bias because computing the global update as the product of the averaged low-rank matrices, i.e., $\Delta_W = (\frac{1}{K}\sum \boldsymbol{B}_k)(\frac{1}{K}\sum \boldsymbol{A}_k)$, is not equivalent to averaging the full-rank updates $\Delta_W^* = \frac{1}{K}\sum \boldsymbol{B}_k \boldsymbol{A}_k$. Specifically, the product of the averaged low-rank matrices $\boldsymbol{B}_k$ and $\boldsymbol{A}_k$ does

not yield the same result as the average of the original weight matrices $\boldsymbol{B}_k \boldsymbol{A}_k$. Consequently, directly aggregating low-rank matrices $\boldsymbol{B}_k$ and $\boldsymbol{A}_k$ may introduce inconsistencies, resulting in suboptimal global updates and slower convergence in FL.

$$
\begin{aligned}
\Delta_W &= \left(\frac{1}{K}\sum_{k=1}^{K} \boldsymbol{B}_k\right)\left(\frac{1}{K}\sum_{k=1}^{K} \boldsymbol{A}_k\right), \\
\Delta_W^* &= \frac{1}{K}\sum_{k=1}^{K} \Delta w_k = \frac{1}{K}\sum_{k=1}^{K} \boldsymbol{B}_k \boldsymbol{A}_k.
\end{aligned}
\tag{8}
$$

Existing FedLoRA variants (Qi et al., 2024) mainly focus on uploading only one of the two matrices ($\boldsymbol{A}$ or $\boldsymbol{B}$) to resolve the mismatch between $\Delta_W$ and $\Delta_W^*$ in Eq.8, but the reduction in uploaded parameters also limits the amount of shared information that can be aggregated. This hinders the aggregation of some shared information. Inspired by FedSA-LoRA (Guo et al., 2024) and our gradient analysis (see Appendix), we observe that the matrices $\boldsymbol{A}_k$ tend to encode more global and stable directions, whereas $\boldsymbol{B}_k$ are more sensitive to client-specific data. To better extract this global information and reduce bias without increasing communication overhead, we propose $FedA^2$-LoRA.

Since $\boldsymbol{A}_k \in \Phi_k$ primarily captures such global directions, we directly aggregate $\boldsymbol{A}_k$ collected from the clients to obtain the global $\boldsymbol{A}$, as shown in Eq.9. This approach follows the same processing method as in the previous FedLoRA approach, where each client uploads its local $\boldsymbol{A}_k$ parameters, which are then aggregated on the server to form the global $\boldsymbol{A}$. This method ensures that the global $\boldsymbol{A}$ reflects the shared global information across clients, without being influenced by the local data distributions. Consequently, the aggregation process is more consistent and unbiased, enabling the model to capture the generalizable patterns better while avoiding overfitting to client-specific data.

$$
\boldsymbol{A} = \frac{1}{K}\sum_{k=1}^{K} \boldsymbol{A}_k
\tag{9}
$$

To effectively integrate the aggregated information from the parameters $\boldsymbol{B}_k$ and $\boldsymbol{A}_k \in \Phi_k$, while preserving global information and still reflecting the unique contribution of each client $k$, we perform aggregation in $\boldsymbol{B}_k \boldsymbol{A}_k$, which is also our goal of optimization. This approach ensures that the federated learning process captures both shared global knowledge and client-specific data without overfitting to individual data distributions, as shown in Eq. 10:

$$
U = \Delta_W^* = \frac{1}{K}\sum_{k=1}^{K} \boldsymbol{B}_k \boldsymbol{A}_k
\tag{10}
$$

By the definitions of Eq. 9 and Eq. 10, we have transformed the multi-objective optimization into a single-objective optimization. The optimization objective can be

*Table 1.* Performance Comparison of Different LoRA Methods on the Heysquad and PandaGPT's Datasets, with Only the **Image** Modality Client Participating in Training.'- -' indicates methods where no training was conducted.Where $\tau$=0.25 and E=1.

| Test Modality | Image | | | | Audio | | | |
|---|---|---|---|---|---|---|---|---|
| Method | $Acc$ | $P_{Bert}$ | $R_{Bert}$ | $F_{Bert}$ | $Acc$ | $P_{Bert}$ | $R_{Bert}$ | $F_{Bert}$ |
| - - | 61.23 | 78.59 | 81.74 | 80.09 | 50.19 | 78.96 | 81.71 | 80.02 |
| LoRA | 64.68 | 81.55 | 81.58 | 81.55 | 52.27 | 81.86 | 79.40 | 80.61 |
| LoRA (II stage) | 66.67 | 84.67 | 82.24 | 83.43 | 55.15 | 83.02 | 80.09 | 81.53 |
| FFA-LoRA | 64.70 | 82.41 | 82.31 | 82.36 | 52.07 | 82.82 | 81.14 | 81.96 |
| FFA-LoRA (II stage) | 70.58 | 85.04 | 80.76 | 82.84 | 55.29 | 81.71 | 82.28 | 81.99 |
| FedSA-LoRA | 68.18 | **87.28** | 76.64 | 81.59 | 54.54 | 83.09 | 78.93 | 80.96 |
| FedSA-LoRA (II stage) | 69.23 | 82.82 | 81.14 | 81.96 | 56.27 | 82.91 | 80.64 | 81.71 |
| FedEx-LoRA | 67.02 | 79.15 | 85.19 | 81.99 | 53.88 | 80.74 | 80.64 | 80.68 |
| FedEx-LoRA (II stage) | 69.40 | 80.75 | **86.91** | 83.72 | 58.33 | 81.28 | **83.02** | 82.13 |
| $FedA^2$-LoRA | 69.02 | 80.66 | 86.85 | 83.63 | 55.56 | 80.75 | 83.00 | 81.86 |
| $FedA^2$-LoRA (II stage) | **72.22** | 82.27 | 86.36 | **84.27** | **58.44** | **85.89** | 80.31 | **83.00** |

written as:

$$\min_B \|\boldsymbol{BA} - U\|_F^2 \qquad (11)$$

The optimal solution $\boldsymbol{B}^* = U\boldsymbol{A}^T[\boldsymbol{AA}^T]^{-1}$ can be obtained by taking the derivative.When $\boldsymbol{AA}^T$ is invertible, the pseudoinverse is equal to the standard inverse $((\boldsymbol{AA}^T)^\dagger = (\boldsymbol{AA}^T)^{-1})$ and is unique, allowing for a straightforward solution. However, when $\boldsymbol{AA}^T$ is non-invertible (or singular), the solution (via the pseudoinverse) is non-unique, and direct (brute-force) solving often leads to an ill-conditioned matrix and numerically unstable results. Therefore, when $\boldsymbol{AA}^T$ is non-invertible, we need to constrain the solution $\boldsymbol{B}$ through regularization (Tikhonov regularization (Groetsch, 1984)), ensuring a unique and stable.Therefore, we can rewrite the optimization objective as follows:

$$\min_B \|\boldsymbol{BA} - U\|_F^2 + \lambda\|\boldsymbol{B}\|_F^2 \qquad (12)$$

In the same manner, by taking the derivative, we can obtain the optimal solution for $B^*$ as $U\boldsymbol{A}^T[\boldsymbol{AA}^T + \lambda I]^{-1}$ when $\boldsymbol{AA}^T$ is non-invertible. By summarizing the results above, we obtain the optimal method for solving $\boldsymbol{B}$ (in Appendix C for details), as follows Eq. 13:

$$\boldsymbol{B} = \begin{cases} U\boldsymbol{A}^T[\boldsymbol{AA}^T + \lambda I]^{-1} & \text{,if } (\boldsymbol{AA}^T)^{-1} \text{ does not exist} \\ U\boldsymbol{A}^T[\boldsymbol{AA}^T]^{-1} & \text{,if } (\boldsymbol{AA}^T)^{-1} \text{ exists} \end{cases} \qquad (13)$$

Where $\lambda$ is a hyperparameter. When the rank $r$ is small, $\boldsymbol{AA}^T$ is typically well-conditioned and the influence of $\lambda$ is minor. As $r$ increases, a smaller $\lambda$ can better restore $\boldsymbol{B}$, but too small a $\lambda$ may lead to numerical instability when inverting $\boldsymbol{AA}^T + \lambda I$ over many rounds. In practice, we find that setting $\lambda = 1$ strikes a good balance between stability and reconstruction accuracy, and ensures that $FedA^2$-LoRA converges reliably in multi-round FL training.We have provided pseudocode in the Appendix A to illustrate the complete process of the model.

## 4. Experiments

In this section, we evaluate the effectiveness of 🌀**UniFLoW** on multimodal question answering using the open Audio QA from HeySQuAD (Wu et al., 2024b) and the open image QA from PandaGPT (Su et al., 2023). Our **UniFLoW** implementation adopts Vicuna (Chiang et al., 2023) as the base LLM and ImageBind (Girdhar et al., 2023) as the modality encoder, with detailed configurations summarized in Table 6 in Appendix D.To demonstrate the substitutability of the base model, we also included the LLAMA-1B (Zhang et al., 2024) model for comparison, as shown in Appendix Table 13.

Beyond multimodal understanding and generation, we further assess the proposed $FedA^2$-LoRA on natural language understanding to verify its generality as a federated LoRA aggregation scheme. For these experiments, we use RoBERTa (Liu et al., 2019) on the GLUE benchmark (Wang et al., 2018), including MNLI, SST-2, and RTE. Our $FedA^2$-LoRA implementation is built on the FederatedScope-LLM library (Kuang et al., 2023). LoRA-based experiments on GLUE are conducted with half-precision for improved efficiency on NVIDIA GeForce RTX 4090 GPUs, while multimodal experiments for **UniFLoW**, rsLoRA, and VeRA are run on NVIDIA A800 GPUs. The main results reported in tables are averaged over multiple runs with mean and standard deviation, and others are obtained from a single run.

### 4.1. UniFlow Multimodal Performance Evaluation

Since datasets Heysquad (Wu et al., 2024b) and PandGPT (Su et al., 2023) are both open-domain QA, traditional evaluation schemes are not well-suited. Therefore, we adopt BERTScore ($P_{Bert}$, $R_{Bert}$, $F_{Bert}$) (Devlin et al., 2019) and token accuracy ($Acc$) (Jiang et al., 2021) as evaluation metrics.We selected recent works, FedLoRA (Wu et al., 2024a), FFA-LoRA (Sun et al., 2024b), and FedSA-LoRA

*Table 2.* Performance Comparison of Different LoRA Methods on the Heysquad and PandaGPT's Datasets, with Only the **Audio** Modality Client Participating in Training.'- -' indicates methods where no training was conducted.Where $\tau$=0.25 and E=1.

| Test Modality | Image | | | | Audio | | | |
|---|---|---|---|---|---|---|---|---|
| Method | $Acc$ | $P_{Bert}$ | $R_{Bert}$ | $F_{Bert}$ | $Acc$ | $P_{Bert}$ | $R_{Bert}$ | $F_{Bert}$ |
| - - | 61.23 | 78.59 | 81.74 | 80.09 | 50.19 | 78.96 | 81.71 | 80.02 |
| LoRA | 57.14 | 77.10 | 79.74 | 78.38 | 54.87 | 76.78 | 83.19 | 79.85 |
| LoRA (II stage) | 62.89 | 79.20 | 81.14 | 80.16 | 54.54 | 80.25 | 82.64 | 81.42 |
| FFA-LoRA | 62.94 | 77.33 | 81.84 | 79.52 | 57.14 | 80.87 | 84.12 | 82.40 |
| FFA-LoRA (II stage) | 63.29 | 81.72 | 81.64 | 81.68 | 55.55 | 83.06 | 83.20 | 83.21 |
| FedSA-LoRA | 63.19 | 78.72 | 83.30 | 80.93 | 53.84 | 80.37 | 80.21 | 80.25 |
| FedSA-LoRA(II stage) | 63.15 | 82.82 | 81.14 | 81.96 | 53.84 | 84.77 | 84.43 | 84.60 |
| FedEx-LoRA | 62.71 | 80.73 | 80.76 | 80.71 | 55.52 | 79.55 | 82.68 | 81.08 |
| FedEx-LoRA(II stage) | 62.96 | 81.39 | 81.72 | 81.44 | 56.80 | 81.31 | 83.89 | 82.58 |
| $FedA^2$-LoRA | 63.00 | **83.62** | 80.30 | 81.92 | 56.63 | 80.32 | 82.89 | 81.57 |
| $FedA^2$-LoRA (II stage) | **63.63** | 81.01 | **83.33** | **82.14** | **58.15** | **85.51** | **85.61** | **85.56** |

*Table 3.* Performance Comparison of Different LoRA Methods on the Heysquad and PandaGPT's Datasets, with the **Image** and **Audio** Modality Client Participating in Training.'- -' indicates methods where no training was conducted.Where $\tau$=0.25 and E=1.

| Test Modality | Image | | | | Audio | | | |
|---|---|---|---|---|---|---|---|---|
| Method | $Acc$ | $P_{Bert}$ | $R_{Bert}$ | $F_{Bert}$ | $Acc$ | $P_{Bert}$ | $R_{Bert}$ | $F_{Bert}$ |
| - - | 61.23 | 78.59 | 81.74 | 80.09 | 50.19 | 78.96 | 81.71 | 80.02 |
| LoRA | 64.89 | **86.13** | 82.22 | 84.11 | 58.33 | 82.42 | **87.75** | 84.97 |
| LoRA (II stage) | 66.21 | 85.75 | 81.57 | 83.60 | 59.63 | 88.64 | 82.44 | 85.42 |
| FFA-LoRA | 63.15 | 85.18 | 79.27 | 82.08 | 57.87 | **91.20** | 77.04 | 83.26 |
| FFA-LoRA (II stage) | 70.58 | 85.61 | 82.07 | 83.80 | 58.43 | 86.18 | 85.86 | 85.82 |
| FedSA-LoRA | 66.29 | 84.22 | 83.15 | 83.68 | 58.33 | 82.32 | 78.93 | 83.67 |
| FedSA-LoRA (II stage) | 68.42 | 84.79 | 83.71 | 84.22 | 60.23 | 84.79 | 83.71 | 84.22 |
| FedEx-LoRA | 66.67 | 81.21 | 85.81 | 83.45 | 58.33 | 83.92 | 83.82 | 83.79 |
| FedEx-LoRA (II stage) | 69.40 | 81.89 | 88.06 | 84.86 | 59.76 | 82.78 | 86.50 | 84.60 |
| $FedA^2$-LoRA | 75.00 | 84.51 | 84.34 | 84.43 | 60.17 | 84.97 | 85.83 | 85.39 |
| $FedA^2$-LoRA (II stage) | **76.19** | 83.44 | **88.08** | **85.69** | **60.90** | 87.15 | 85.49 | **86.31** |

(Guo et al., 2024), for comparison. Since the model is a pre-trained large model, only a limited communication rounds are required. In these experiments, we set the communication rounds to 10, with 10 participating clients per modality. Each client is provided with 2,000 samples for training and 200 samples for testing. The detailed descriptions of the datasets and evaluation metrics are provided in Appendix H.

As shown in Tables 2 and 1, training on data from a single modality enhances the base model's QA ability. Moreover, Table 3 demonstrates that the performance of clients improves even when modality inconsistencies exist across clients. However, as indicated in Table 2, directly applying LoRA in FL may lead to performance degradation. From Tables 1, 2, and 3, we observe that our II stage training strategy provides an effective approach for mitigating modality differences in FL.

### 4.2. $FedA^2$-LoRA Performance Evaluation

As shown in Table 4, the proposed $FedA^2$-LoRA aggregation strategy consistently improves the performance of

different LoRA variants on the GLUE benchmark. In particular, $FedA^2$-rsLoRA achieves the highest average accuracy of 90.68, a clear improvement over the original rsLoRA (89.16). Similar gains are observed for both LoRA and VeRA, indicating that $FedA^2$-LoRA can generally enhance the effectiveness of existing federated LoRA methods. These results suggest that $FedA^2$-LoRA not only alleviates the aggregation inconsistency typically encountered in federated LoRA, but also improves model stability and generalization across diverse tasks, without introducing additional communication overhead. In these experiments, we use 3 participating clients and run 1000 rounds to evaluate the aggregation strategy under a practical FL setup.Since the dataset is highly sensitive to the learning rate—making convergence difficult in most cases—we achieved superior results by adjusting the learning rate.During the early stage of federated training, we observe that the inversion results in our method can be unstable and may be largely overwritten due to limited training precision. To ensure a fair comparison under identical precision conditions, we therefore adopt a two-stage training strategy. Specifically, we use FedLoRA during the first 200 communication rounds and switch to $FedA^2$-LoRA for the

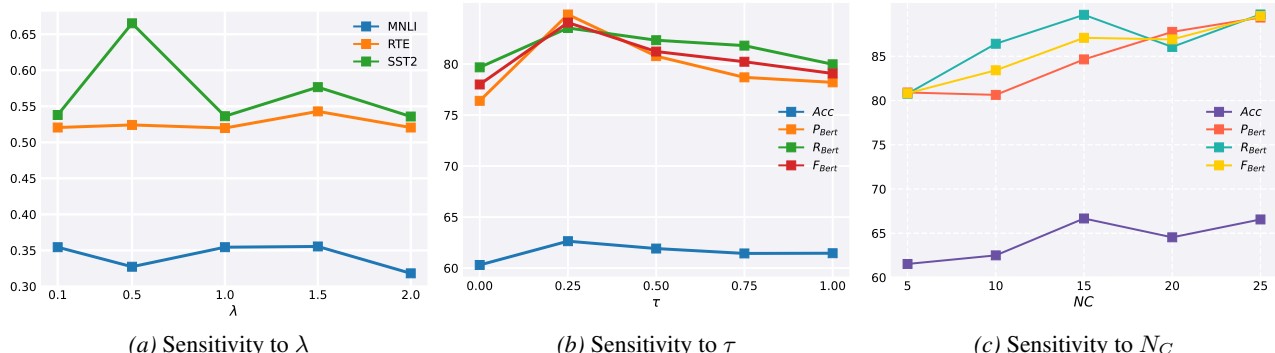

| | | | |
|---|---|---|---|
| *(a)* Sensitivity to $\lambda$ | *(b)* Sensitivity to $\tau$ | *(c)* Sensitivity to $N_C$ |

*Figure 3.* Sensitivity analysis of the performance with respect to different hyperparameters. (a) Sensitivity to $\lambda$, showing the performance variation across different tasks (MNLI, RTE, and SST2). (b) and (c) Sensitivity to $\tau$ and $N_C$ (the Number of Clients), demonstrating the impact on $Acc$, $P_{Bert}$, $R_{Bert}$, and $F_{Bert}$.

*Table 4.* Performance of different methods on the GLUE benchmark. For all tasks, we report accuracy evaluated across 3 runs with mean and standard deviation.

| | Method | MNLI | RTE | SST2 | QQP | Avg. |
|---|---|---|---|---|---|---|
| LoRA | LoRA | $88.71_{\pm 0.09}$ | $87.49_{\pm 0.15}$ | $95.16_{\pm 0.09}$ | $85.33_{\pm 1.33}$ | 89.17 |
| | FFA-LoRA | $88.83_{\pm 0.02}$ | $86.08_{\pm 1.16}$ | $94.95_{\pm 0.04}$ | $86.71_{\pm 0.07}$ | 89.14 |
| | FedDPA-LoRA | $88.99_{\pm 0.06}$ | $87.44_{\pm 0.13}$ | $95.50_{\pm 0.06}$ | $85.73_{\pm 1.73}$ | 89.41 |
| | FedSA-LoRA | $90.18_{\pm 0.02}$ | $87.93_{\pm 0.11}$ | $96.00_{\pm 0.04}$ | $87.48_{\pm 0.22}$ | 90.39 |
| | $FedA^2$-LoRA | $90.27_{\pm 0.54}$ | $88.05_{\pm 0.59}$ | $96.13_{\pm 0.26}$ | $87.53_{\pm 0.12}$ | 90.50 |
| rsLoRA | rsLoRA | $88.91_{\pm 0.15}$ | $85.99_{\pm 0.34}$ | $95.02_{\pm 0.24}$ | $86.73_{\pm 0.98}$ | 89.16 |
| | FFA-rsLoRA | $89.21_{\pm 0.11}$ | $85.24_{\pm 0.21}$ | $95.42_{\pm 0.17}$ | $86.93_{\pm 1.18}$ | 89.20 |
| | FedDPA-rsLoRA | $89.34_{\pm 0.11}$ | $86.26_{\pm 0.11}$ | $95.56_{\pm 0.21}$ | $86.81_{\pm 0.53}$ | 89.49 |
| | FedSA-rsLoRA | $90.35_{\pm 0.11}$ | $88.00_{\pm 0.10}$ | $95.78_{\pm 0.08}$ | $87.97_{\pm 0.16}$ | 90.52 |
| | $FedA^2$-rsLoRA | $90.37_{\pm 0.58}$ | $88.42_{\pm 0.17}$ | $95.86_{\pm 0.29}$ | $88.04_{\pm 0.62}$ | 90.68 |
| VeRA | VeRA | $85.54_{\pm 0.10}$ | $86.31_{\pm 0.12}$ | $93.53_{\pm 0.13}$ | $82.07_{\pm 0.35}$ | 86.86 |
| | FFA-VeRA | $86.63_{\pm 0.13}$ | $83.54_{\pm 0.59}$ | $93.44_{\pm 0.05}$ | $82.23_{\pm 0.07}$ | 86.46 |
| | FedDPA-VeRA | $86.74_{\pm 0.11}$ | $86.12_{\pm 0.12}$ | $93.61_{\pm 0.32}$ | $82.11_{\pm 0.41}$ | 87.14 |
| | FedSA-VeRA | $87.21_{\pm 0.10}$ | $87.83_{\pm 0.09}$ | $93.68_{\pm 0.07}$ | $82.56_{\pm 0.05}$ | 87.82 |
| | $FedA^2$-$VeLoRA$ | $87.29_{\pm 0.13}$ | $88.00_{\pm 0.32}$ | $94.08_{\pm 0.14}$ | $82.70_{\pm 0.22}$ | 88.02 |

remaining 800 rounds. All other experimental settings are kept unchanged across methods.

### 4.3. Ablation Study

**Sensitivity to $\lambda$.** As shown in Figure 3a, when $FedA^2$-LoRA restores $\boldsymbol{B}$, it does so by inverting it. A hyperparameter $\lambda$ is introduced during **regularization**. Although mathematically, smaller values of $\lambda$ are preferable, it grows exponentially during federated learning iterations. As a result, an excessively small $\lambda$ can lead to memory overflows. To address this, we set $\lambda = 1$ and conducted a sensitivity analysis. Our results show that $FedA^2$-LoRA is not significantly affected by $\lambda$, likely because the number of regularizations decreases when the matrix $r$ of $\boldsymbol{B}$ and $\boldsymbol{A}$ is small. This was verified testing on MNLI, RTE, and SST2.

**Sensitivity to $\tau$.** As shown in Figure 3b, performance reaches its peak at $\tau = 0.25$. This may be attributed to the parameter ratio between the ImageBind and Vicuna mod-els. At $\tau = 0$, only the base model is fine-tuned, with the encoder left unchanged. Conversely, at $\tau = 1$, only the encoder is fine-tuned, while the base model remains fixed. This observation indirectly highlights the effectiveness of two-stage training, where both components contribute to achieving optimal performance.

**Sensitivity to $N_C$.** As shown in Figure 3c, the performance of the model varies with the number of clients (denoted as $N_C$). The results indicate that as the number of clients increases from 5 to 25, the accuracy ($Acc$) and other metrics (such as Precision, Recall, and F1 score) improve significantly, demonstrating the positive impact of adding more clients in FL. This suggests that increasing the number of participating clients helps enhance the **UniFLoW** to generalize, as it learns from a more diverse set of data sources. However, the improvement in performance starts to plateau beyond a certain point, indicating diminishing returns with a higher number of clients. These observations highlight the importance of optimizing the number

of clients to balance performance gains with computational and communication efficiency.

## 5. Conclusion

We introduced 🌀**UniFLoW**, a unified federated framework for multimodal large language model fine-tuning. Our framework effectively addresses critical challenges in federated learning, including modality heterogeneity, the risk of overfitting with limited private data, and aggregation bias in LoRA-based fine-tuning. Through our novel and efficient scheme $FedA^2$-LoRA and our stage training strategy, **UniFLoW** demonstrated its ability to effectively utilize fragmented multimodal data without incurring additional communication costs. Our extensive experiments across speech, image, and text modalities have shown that **UniFLoW** not only achieves state-of-the-art performance but also maintains consistency and efficiency in aggregation. The II stage training approach we implemented proved crucial in mitigating overfitting and ensuring the proper integration of both modality-specific and content-specific information. We believe **UniFLoW** offers a promising approach to unlocking the full potential of private multimodal data, especially as public resources for large-model training become increasingly scarce. Future research could explore its application to more modalities and its scalability with a larger number of clients.

## Acknowledgments

This work was supported by the Guangdong S&T Program 2024B0101040005, National Natural Science Foundation of China (Grant No. T2125006 and No. 42401415), Shenzhen Science and Technology Program (KCXFZ20240903093759004 and KJZD20230923115106012), and Guangdong S&T Program 2025B0101080001.

## Impact Statement

This paper presents work whose goal is to advance the field of Machine Learning. There are many potential societal consequences of our work, none which we feel must be specifically highlighted here.

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

# UniFLoW: Universal Multi-Modal Federated LoRA Fine-Tuning Framework with Analytical Aggregation (Supplementary material)

## Table of Contents in Appendix

## A. Pseudocode for UniFLoW

---

**Algorithm 1** UniFLOW Framework with $FedA^2$-LoRA Aggregation

---

**Require:** Dataset $\mathcal{D}_k$ for each client $k$, Modality Encoders $f_m$, LLM $\mathcal{F}$, Communication rounds $T$, Stage switch threshold $\tau$, Regularization coefficient $\lambda$, Learning rate $\eta$.
**Ensure:** Global LoRA parameters $\Phi_{global} = \{A, B\}$.
 1: **Server Initialization:** Initialize LoRA matrices $A^{(0)}, B^{(0)}$ for Encoders and LLM.
 2: **for** $t = 0$ to $T - 1$ **do**
 3:     Select a subset of clients $\mathcal{S}_t$.
 4:     **for** each client $k \in \mathcal{S}_t$ **in parallel do**
 5:         $\Delta\Phi_k^{(t+1)} \leftarrow \text{ClientUpdate}(k, A^{(t)}, B^{(t)}, t)$
 6:     **end for**
 7:     **Server Aggregation ($FedA^2$-LoRA):**
 8:     Collect $\{A_k, B_k\}$ from clients.
 9:     *// Aggregate A via averaging (Eq. 9)*
10:     $A^{(t+1)} = \frac{1}{|\mathcal{S}_t|} \sum_{k \in \mathcal{S}_t} A_k$
11:     *// Compute target global update U (Eq. 10)*
12:     $U = \frac{1}{|\mathcal{S}_t|} \sum_{k \in \mathcal{S}_t} B_k A_k$
13:     *// Analytically solve for B using Ridge Regression (Eq. 13)*
14:     **if** $(A^{(t+1)}(A^{(t+1)})^T)$ is invertible **then**
15:         $B^{(t+1)} = U(A^{(t+1)})^T[A^{(t+1)}(A^{(t+1)})^T]^{-1}$
16:     **else**
17:         $B^{(t+1)} = U(A^{(t+1)})^T[A^{(t+1)}(A^{(t+1)})^T + \lambda I]^{-1}$
18:     **end if**
19:     Broadcast $\Phi_{global}^{(t+1)} = \{A^{(t+1)}, B^{(t+1)}\}$ to clients.
20: **end for**
21: **function ClientUpdate**$(k, A, B, t)$
22:     Receive global parameters $A, B$.
23:     Initialize local params: $A_k \leftarrow A, B_k \leftarrow B$.
24:     *// UniFLoW II-Stage Training Strategy (Eq. 7)*
25:     **if** $t < \tau$ **then**
26:         **Stage I:** Freeze LLM parameters. Train Modality Encoder LoRA ($\Phi_E$).
27:         $\mathcal{L} \leftarrow \text{ComputeLoss}(\mathcal{D}_k, \Phi_E)$
28:         Update $\Phi_E \leftarrow \Phi_E - \eta\nabla_{\Phi_E}\mathcal{L}$
29:     **else**
30:         **Stage II:** Freeze Encoder parameters. Train LLM LoRA ($\Phi_{LLM}$).
31:         $\mathcal{L} \leftarrow \text{ComputeLoss}(\mathcal{D}_k, \Phi_{LLM})$
32:         Update $\Phi_{LLM} \leftarrow \Phi_{LLM} - \eta\nabla_{\Phi_{LLM}}\mathcal{L}$
33:     **end if**
        **return**    Updated local LoRA matrices $\{A_k, B_k\}$.
34: **end function**

---

## B. Proof of Lemma B.1

**Lemma B.1.** *the $A$ matrices tend to capture more global and stable directions than the $B$ matrices.*

*Proof.* Inspired by (Guo et al., 2024), we provide the following proof: since the computations of B and A are independent, we can fix one to optimize the other. First, we consider fine-tuning $A$ with fixed $B = \mathbb{B}$. The loss function becomes:

$$\mathcal{L} = \mathbb{E}_{(X_{n_k}, \mathcal{A}_{y_{n_k}})}[\|\mathcal{A}_{y_{n_k}} - (W_0 + \mathbb{B}A)X_{n_k}\|_2^2]. \tag{14}$$

Then, the gradients of Eq. (17) w.r.t. $A$ is:

$$
\begin{aligned}
\frac{\partial \mathcal{L}}{\partial A} &= \frac{\partial \mathbb{E}_{(X_{n_k}, \mathcal{A}_{y_{n_k}})}[\|\mathcal{A}_{y_{n_k}} - (W_0 + \mathbb{B}A)X_{n_k}\|_2^2]}{\partial A} \\
&= \frac{\partial \mathbb{E}[\|W_t X_{n_k} - (W_0 + \mathbb{B}A)X_{n_k}\|_2^2]}{\partial A} \\
&= \frac{\partial \mathbb{E}[\|(W_0 + \Delta_W)X_{n_k} - (W_0 + B\mathbb{A})X_{n_k}\|_2^2]}{\partial A} \\
&= \frac{\partial \mathbb{E}[\|(\Delta_W - \mathbb{B}A)X_{n_k}\|_2^2]}{\partial A} \\
&= \mathbb{E}[2\mathbb{B}^T[(\Delta_W - \mathbb{B}A)X_{n_k}]X_{n_k}^T]
\end{aligned}
\tag{15}
$$

To obtain the optimal $A^*$, we set Eq. (15) to zero, which means:

$$
\begin{aligned}
\mathbb{E}[2\mathbb{B}^T[(\Delta_W - \mathbb{B}A)X_{n_k}]X_{n_k}^T] &= 0 \\
2\mathbb{B}^T \Delta_W \mathbb{E}[X_{n_k} X_{n_k}^T] - 2\mathbb{B}^T \mathbb{B}A \mathbb{E}[X_{n_k} X_{n_k}^T] &= 0 \\
\mathbb{B}^T \mathbb{B}A \mathbb{E}[X_{n_k} X_{n_k}^T] &= \mathbb{B}^T \Delta_W \mathbb{E}[X_{n_k} X_{n_k}^T] \\
A &= \mathbb{B}^\dagger \Delta_W.
\end{aligned}
\tag{16}
$$

Thus, we obtain $A^* = \mathbb{B}^\dagger \Delta_W$.

Then, we consider fine-tuning $B$ while freezing $A = \mathbb{A}$. We abstract the loss function as:

$$
\mathcal{L} = \mathbb{E}_{(X_{n_k}, \mathcal{A}_{y_{n_k}})}[\|\mathcal{A}_{y_{n_k}} - (W_0 + BA)X_{n_k}\|_2^2].
\tag{17}
$$

Then, the gradient of Eq. (17) w.r.t. $B$ is:

$$
\begin{aligned}
\frac{\partial \mathcal{L}}{\partial B} &= \frac{\partial \mathbb{E}_{(X_{n_k}, \mathcal{A}_{y_{n_k}})}[\|\mathcal{A}_{y_{n_k}} - (W_0 + B\mathbb{A})X_{n_k}\|_2^2]}{\partial B} \\
&= \frac{\partial \mathbb{E}[\|\mathcal{A}_{y_{n_k}} - (W_0 + B\mathbb{A})X_{n_k}\|_2^2]}{\partial B} \\
&= \frac{\partial \mathbb{E}[\|(W_0 + \Delta_W)X_{n_k} - (W_0 + B\mathbb{A})X_{n_k}\|_2^2]}{\partial B} \\
&= \frac{\partial \mathbb{E}[\|(\Delta_W - \mathbb{B}A)X_{n_k}\|_2^2]}{\partial B} \\
&= \mathbb{E}[2[(\Delta_W - B\mathbb{A})X_{n_k}](-X_{n_k}^T \mathbb{A}^T)] \\
&= \mathbb{E}[2(B\mathbb{A} - \Delta_W)X_{n_k} X_{n_k}^T \mathbb{A}^T].
\end{aligned}
\tag{18}
$$

To obtain the optimal $B^*$, we set Eq. (18) to zero, which means:

$$
\begin{aligned}
\mathbb{E}[2(B\mathbb{A} - \Delta_W)X_{n_k} X_{n_k}^T \mathbb{A}^T] &= 0 \\
2B\mathbb{A}\mathbb{E}[X_{n_k} X_{n_k}^T]\mathbb{A}^T - 2\Delta_W \mathbb{E}[X_{n_k} X_{n_k}^T]\mathbb{A}^T &= 0 \\
2B\mathbb{A}\mathbb{E}[X_{n_k} X_{n_k}^T]\mathbb{A}^T - 2\Delta_W \mathbb{E}[X_{n_k} X_{n_k}^T]\mathbb{A}^T &= 0 \\
B\mathbb{A}\mathbb{E}[X_{n_k} X_{n_k}^T]\mathbb{A}^T &= \Delta_W \mathbb{E}[X_{n_k} X_{n_k}^T]\mathbb{A}^T \\
B &= \Delta_W \mathbb{E}[X_{n_k} X_{n_k}^T]\mathbb{A}^T (\mathbb{A}\mathbb{E}[X_{n_k} X_{n_k}^T]\mathbb{A}^T)^{-1}.
\end{aligned}
\tag{19}
$$

Therefore, we obtain $B^* = \Delta_W \mathbb{E}[X_{n_k} X_{n_k}^T]\mathbb{A}^T (\mathbb{A}\mathbb{E}[X_{n_k} X_{n_k}^T]\mathbb{A}^T)^{-1}$.

Therefore, we can derive the following relationship between $A^*$ and $B^*$:

$$
B^* = \mathbb{B}A^* \mathbb{E}[X_{n_k} X_{n_k}^T]\mathbb{A}^T (\mathbb{A}\mathbb{E}[X_{n_k} X_{n_k}^T]\mathbb{A}^T)^{-1}
\tag{20}
$$

Where $\mathbb{A}$ and $\mathbb{B}$ are Gaussian random initialization matrices.

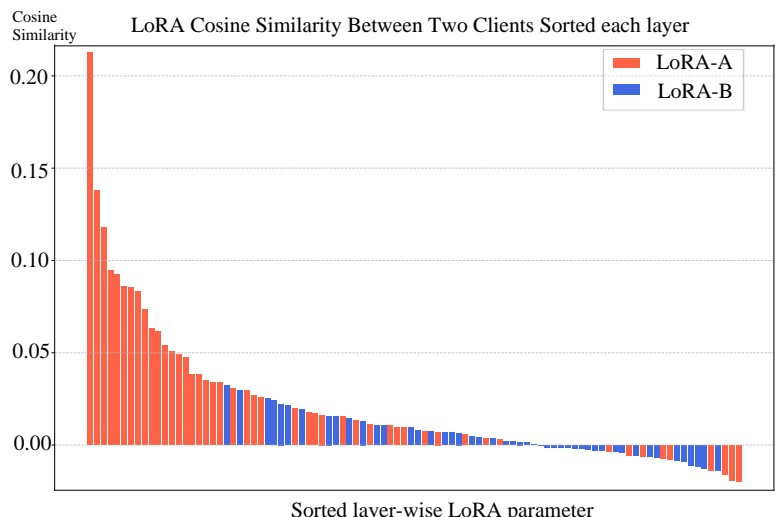

*Figure 4.* Similarity between LoRA-A and LoRA-B at each layer of the two clients

From Eq. 20, we know that $A^*$ only contains information unrelated to the data, which we consider to be general information. $B^*$ can be written as a function of $A^*$ and $X_{n_k}$, meaning it can capture both general and data information. Since A and B usually have the same rank, $A^*$ contains more general information when acquiring the same amount of information.

Meanwhile, since $B^*$ can be written as a function of $A^*$ and $X_{n_k}$, it indicates that $B$ also contains general information, making the aggregation of $B$ necessary. Therefore, we propose $FedA^2$-LoRA to fully capture global information by recovering $\Delta_W^* = \frac{1}{K}\sum_{k=1}^{K} \boldsymbol{B}_k\boldsymbol{A}_k$. $\qquad\square$

## C. PROOF OF Eq. 13

$FedA^2$-LoRA primarily aims to mitigate the problem of $\Delta_W = \left(\frac{1}{K}\sum_{k=1}^{K} \boldsymbol{B}_k\right)\left(\frac{1}{K}\sum_{k=1}^{K} \boldsymbol{A}_k\right) \neq \Delta_W^* = \frac{1}{K}\sum_{k=1}^{K} \boldsymbol{B}_k\boldsymbol{A}_k$. Therefore, we need to design B and A to restore u as much as possible, and thus we can express the optimization objective as Eq. 21.

$$\min_B \|\boldsymbol{B}\boldsymbol{A} - U\|_F^2 \tag{21}$$

As shown in Appendix C, $A$ more easily captures general information, while $B$ contains both general and personalized information. Therefore, we consider first solving for $\boldsymbol{A}$ using a weighted average method, as shown in Eq. 9. Then, we only need to design the optimization function to solve for $\boldsymbol{B}$. The optimization function is as follows:

$$f(\boldsymbol{B}) = \|\boldsymbol{B}\boldsymbol{A} - U\|_F^2 \tag{22}$$

We compute the derivative of the objective function with respect to $\boldsymbol{B}$, we have

$$
\begin{aligned}
\frac{\partial f}{\partial B} &= \frac{\partial \|\boldsymbol{B}\boldsymbol{A} - U\|_F^2}{\partial B} \\
&= \frac{\partial}{\partial B}\mathrm{tr}((\boldsymbol{B}\boldsymbol{A} - U)^T(\boldsymbol{B}\boldsymbol{A} - U)) \\
&= \frac{\partial}{\partial B}\mathrm{tr}(\boldsymbol{A}^T\boldsymbol{B}^T\boldsymbol{B}\boldsymbol{A} - 2U^T\boldsymbol{B}\boldsymbol{A} + U^TU) \\
&= 2\boldsymbol{B}\boldsymbol{A}\boldsymbol{A}^T - 2U\boldsymbol{A}^T
\end{aligned}
\tag{23}
$$

To obtain the optimal $\boldsymbol{B}$, we set Eq. 23 to zero, which means:

$$
\begin{aligned}
2\boldsymbol{B}\boldsymbol{A}\boldsymbol{A}^T - 2U\boldsymbol{A}^T &= 0 \\
\boldsymbol{B}\boldsymbol{A}\boldsymbol{A}^T &= U\boldsymbol{A}^T \\
\boldsymbol{B}^* &= U\boldsymbol{A}^T(\boldsymbol{A}\boldsymbol{A}^T)^{\dagger}
\end{aligned}
\tag{24}
$$

When $\boldsymbol{A}\boldsymbol{A}^T$ is invertible, the pseudoinverse is equal to the standard inverse ($(\boldsymbol{A}\boldsymbol{A}^T)^{\dagger} = (\boldsymbol{A}\boldsymbol{A}^T)^{-1}$) and is unique, allowing for a straightforward solution. However, when $\boldsymbol{A}\boldsymbol{A}^T$ is non-invertible (or singular), the solution (via the pseudoinverse) is non-unique, and direct (brute-force) solving often leads to an ill-conditioned matrix and numerically unstable results. Therefore, when $\boldsymbol{A}\boldsymbol{A}^T$ is non-invertible, we need to constrain the solution $\boldsymbol{B}$ through regularization (Tikhonov regularization), ensuring a unique and stable.Therefore, we can rewrite the optimization objective as follows:

$$
\min_B \|\boldsymbol{B}\boldsymbol{A} - U\|_F^2 + \lambda\|\boldsymbol{B}\|_F^2
\tag{25}
$$

By rewriting the optimization function we can obtain :

$$
F(\boldsymbol{B}) = |\boldsymbol{B}\boldsymbol{A} - U\|_F^2 + \lambda\|\boldsymbol{B}\|_F^2
\tag{26}
$$

We compute the derivative of the objective function with respect to $\boldsymbol{B}$, we have

$$
\begin{aligned}
\frac{\partial F}{\partial B} &= \frac{\partial[\|\boldsymbol{B}\boldsymbol{A} - U\|_F^2 + \lambda\|\boldsymbol{B}\|_F^2]}{\partial B} \\
&= \frac{\partial}{\partial B}\mathrm{tr}((\boldsymbol{B}\boldsymbol{A} - U)^T(\boldsymbol{B}\boldsymbol{A} - U)) + \lambda\frac{\partial}{\partial B}\mathrm{tr}(\boldsymbol{B}^T\boldsymbol{B}) \\
&= \frac{\partial}{\partial B}\mathrm{tr}(\boldsymbol{A}^T\boldsymbol{B}^T\boldsymbol{B}\boldsymbol{A} - 2U^T\boldsymbol{B}\boldsymbol{A} + U^TU + \lambda\boldsymbol{B}^T\boldsymbol{B}) \\
&= 2\boldsymbol{B}\boldsymbol{A}\boldsymbol{A}^T - 2U\boldsymbol{A}^T + 2\lambda\boldsymbol{B}
\end{aligned}
\tag{27}
$$

To obtain the optimal $\boldsymbol{B}$, we set Eq. 27 to zero, which means:

$$
\begin{aligned}
2\boldsymbol{B}\boldsymbol{A}\boldsymbol{A}^T - 2U\boldsymbol{A}^T + 2\lambda\boldsymbol{B} &= 0 \\
\boldsymbol{B}\boldsymbol{A}\boldsymbol{A}^T + \lambda\boldsymbol{B} &= U\boldsymbol{A}^T \\
\boldsymbol{B}(\boldsymbol{A}\boldsymbol{A}^T + \lambda I) &= U\boldsymbol{A}^T
\end{aligned}
\tag{28}
$$

The $\boldsymbol{A}\boldsymbol{A}^T$ is Symmetric Positive Semi-definite (PSD), because the matrix $\boldsymbol{A}\boldsymbol{A}^T$ is a Gram matrix formed by the product of a matrix and its transpose. For any non-zero vector $v$, the quadratic form $v^T(\boldsymbol{A}\boldsymbol{A}^T)v$ can be rewritten as:

$$
v^T(\boldsymbol{A}\boldsymbol{A}^T)v = (\boldsymbol{A}^Tv)^T(\boldsymbol{A}^Tv) = \|\boldsymbol{A}^Tv\|_2^2 \geq 0
$$

Since the squared $L_2$ norm is always non-negative, $\boldsymbol{A}\boldsymbol{A}^T$ is PSD.

And $\boldsymbol{A}\boldsymbol{A}^T + \lambda I$ is Positive Definite (PD) For any non-zero vector $v$, we examine the quadratic form $v^T M v$:

$$
v^T(\boldsymbol{A}\boldsymbol{A}^T + \lambda I)v = v^T(\boldsymbol{A}\boldsymbol{A}^T)v + v^T(\lambda I)v
$$

$$
= \underbrace{\|\boldsymbol{A}^Tv\|_2^2}_{\geq 0} + \underbrace{\lambda\|v\|_2^2}_{>0 \text{ (since } \lambda>0)}
$$

Since $v^T(\boldsymbol{A}\boldsymbol{A}^T)v \geq 0$ and, critically, $\lambda\|v\|_2^2 > 0$ (for $v \neq 0$ and $\lambda > 0$), the sum is strictly positive:

$$
v^T(\boldsymbol{A}\boldsymbol{A}^T + \lambda I)v > 0
$$

Since the quadratic form $v^T(\boldsymbol{A}\boldsymbol{A}^T + \lambda I)v$ is strictly positive for all non-zero vectors $v$, the matrix $\boldsymbol{A}\boldsymbol{A}^T + \lambda I$ is Positive Definite (PD). All positive definite matrices are non-singular and therefore invertible.

*Table 5.* Summary of key notations.

| Symbol | Description |
|---|---|
| $\lambda$ | Regularization hyperparameter, controls the penalty term to prevent overfitting. Sensitive to performance. |
| $\tau$ | The threshold for two-stage training. It determines when to switch from fine-tuning the modality encoders for fine-tuning the base LLM. |
| $N_C$ | The number of participating clients in FL. Affects accuracy, precision, and recall. More clients improve performance, but diminishing returns occur after a certain point. |
| $W_0$ | Frozen parameters of the pre-trained model. |
| $\alpha$ | Scaling factor that controls the degree of LoRA injection. |
| $\boldsymbol{B}, \boldsymbol{A}$ | Trainable parameters used in LoRA adaptation. |
| $Z$ | The features of modality data after processing by the encoder. |
| $\Phi_e$ | The set of trainable parameters for LoRA adjustments in the encoders. |
| $X_{nk}$ | The final input feature for the LLM, combining modality features and question data. |
| $e_{\text{BOS}}$ | Begin-of-sequence token, used to signal the start of a sequence. |
| $P_{\text{before}}$ | Pre-tokenized or pre-processed input, used to initialize the sequence. "Start of the conversation."+"### Human: ⟨modality⟩" |
| $\mathcal{X}_{m_{nk}}$ | Features from modality $m$, processed by the modality-specific encoder. |
| $P_{\text{gap}}$ | Token or placeholder used for bridging different modality features. "⟨\modality⟩ "+"Now, here's the question based on the above features:" |
| $Q_{nk}$ | Question part of the input for tasks like question answering (QA). |
| $P_{\text{after}}$ | Post-processed output or filler token after modality features are processed. "Based on the above input and question, please provide your response." |

*Table 6.* In our 🌊**UniFLoW** system, only 1% of the parameters require updating during fine-tuning, ensuring high efficiency and reduced communication overhead.

| | Encoder | | Encoder LoRA | | LLM | | LLM LoRA | |
|---|---|---|---|---|---|---|---|---|
| | Name | Param | Name | Param | Name | Param | Name | Param |
| **Text** | — | — | — | — | | | | |
| **Image** | ImageBind | 1.2B❄ | $FedA^2$-$LoRA$ | 6M🔥 | $Vicuna$ | 7B❄ | $FedA^2$-$LoRA$ | 33M🔥 |
| **Audio** | | | | | | | | |

Therefore, when $\boldsymbol{A}\boldsymbol{A}^T$ is irreversible, the optimal solution for $B^*$ is as follows:

$$\boldsymbol{B}^* = U\boldsymbol{A}^T(\boldsymbol{A}\boldsymbol{A}^T + \lambda I)^{-1} \tag{29}$$

Therefore, when $\boldsymbol{A}\boldsymbol{A}^T$ is both reversible and irreversible, the optimal solution for $\boldsymbol{B}$, which is the $\boldsymbol{B}$ returned by our server, is as follows:

$$\boldsymbol{B} = \begin{cases} U\boldsymbol{A}^T[\boldsymbol{A}\boldsymbol{A}^T + \lambda I]^{-1} & \text{, if } (\boldsymbol{A}\boldsymbol{A}^T)^{-1} \text{ does not exist} \\ U\boldsymbol{A}^T[\boldsymbol{A}\boldsymbol{A}^T]^{-1} & \text{, if } (\boldsymbol{A}\boldsymbol{A}^T)^{-1} \text{ exist} \end{cases} \tag{30}$$

## D. Notations

Table 5 summarizes the notations appearing in this paper. Table 6 illustrates the parameter distribution in the UniFLoW system, highlighting the efficiency of fine-tuning with minimal updates.

## E. Further analytical experiments

### E.1. Cost Analysis

From Table 7, we can see that $FedA^2 - LoRA$ remains highly cost-effective in terms of overhead: the number of trainable parameters and the per-round communicated parameters are the same as standard LoRA (1.83M / 0.78M) and are much lower than FedEx-LoRA, which requires uploading the full model (25.74M). Therefore, it does not introduce additional

communication or memory pressure. Although we introduce a closed-form aggregation step with computational complexity $O(Krd^2)$, this operation is executed only on the server side and remains manageable under a small LoRA rank $r$. In practice, this small extra computational cost yields more stable consistent aggregation and better federated performance.

*Table 7.* Time and space costs for each method on the RTE and QNLI tasks. Where $K$ represents the number of clients, the weight dimension of $W$ is $d \times d$, and the dimensions of B and A are $d \times r$ and $r \times d$, respectively. Here, is typically $\{1024, 4096\}$, and is typically $\{4, 8, 16\}$.

| | # Trainable Parm. | # Per-round Communicated Parm. | # Computational Complexity. |
|---|---|---|---|
| LoRA | 1.83M | 0.78M | $O(Krd)$ |
| FFA-LoRA | 1.44M | 0.39M | $O(Krd)$ |
| FedDPA-LoRA | 2.62M | 0.78M | $O(Krd)$ |
| FedSA-LoRA | 1.83M | 0.39M | $O(Krd)$ |
| FedEx-LoRA | 1.83M | 25.74M | $O(Krd^2)$ |
| $FedA^2 - LoRA$ | 1.83M | 0.78M | $O(Krd^2)$ |

## E.2. Other indicators

Table 8 shows that $FedA^2 - LoRA$ consistently improves performance across all metrics—including BLEU, ROUGE-L, and METEOR—while maintaining strong results on both image and audio quality assurance. Compared with standard LoRA and other federated LoRA variants, our method achieves the highest or near-highest scores across both modalities. Notably, under the Stage-II setting, $FedA^2 - LoRA$ achieves:

These results demonstrate that our analytical aggregation method not only improves overall accuracy but also enhances output fluency (BLEU), semantic coverage (ROUGE-L), and paraphrase robustness (METEOR). The improvements remain consistent across different modalities, highlighting the effectiveness of FedA²-LoRA in multimodal federated learning scenarios.

## E.3. Performance Under the same communication cost

As shown in Table 9, under the same communication cost (i.e., comparable numbers of communication rounds and per-round transmitted parameters), our proposed $FedA^2 - LoRA$ achieves the best overall performance across both image and audio modalities. For the *image* modality, $FedA^2 - LoRA$ attains 76.92 Acc and 85.93 $F_{\text{BERT}}$, which is competitive with or superior to all baselines while using only half the communication rounds of FFA-LoRA and FedSA-LoRA. For the *audio* modality, $FedA^2 - LoRA$ yields the highest Acc (78.84) and the best $F_{\text{BERT}}$ (85.94), outperforming all competing methods by a clear margin. These results demonstrate that the proposed analytical aggregation not only preserves the communication efficiency of standard LoRA, but also delivers consistently stronger multimodal performance under realistic federated constraints.

## E.4. heterogeneity-aware mechanisms

To better reflect realistic heterogeneous-client scenarios, we further evaluate a heterogeneity-aware variant of **UniFLoW**. In this setting, different clients adopt different LoRA ranks according to their local data scale and computational budget we simulate five clients whose encoders use ranks r=4 or r=8, and we employ zero-padding to handle heterogeneous sequence lengths. As reported in Table 10, the proposed $FedA^2 - LoRA(zero - padding)$ consistently outperforms standard $LoRA(zero - padding)$ on both modalities: for images, it improves Acc and $F_{\text{BERT}}$ from 61.64/82.34 to 61.92/84.46, and for audio from 56.27/82.22 to 57.93/83.46. These results demonstrate that our analytical aggregation remains effective when LoRA ranks and data sizes vary across clients, and that UniFLoW naturally extends to heterogeneity-aware configurations without sacrificing performance.

## E.5. The performance of $FedA^2 - LoRA$ on Image and Audio modalities

As summarized in Table 11, $FedA^2 - LoRA$ consistently achieves the best overall performance among all federated LoRA baselines on both *image* and *audio* modalities. In particular, it substantially improves accuracy and $F_{\text{BERT}}$ over standard LoRA, FFA-LoRA, FedSA-LoRA, and FedEx-LoRA, while maintaining the same communication budget. These results

*Table 8.* Performance Comparison of Different LoRA Methods on the Heysquad and PandaGPT's Datasets, with the **Image** and **Audio** Modality Client Participating in Training.'- -' indicates methods where no training was conducted.

| Test Modality | Image | | | Audio | | |
|---|---|---|---|---|---|---|
| Method | $B_{LEU}$ | $R_{OUGE-L}$ | $M_{ETEOR}$ | $B_{LEU}$ | $R_{OUGE-L}$ | $M_{ETEOR}$ |
| - - | 31.62 | 50.14 | 50.42 | 29.94 | 53.33 | 45.39 |
| LoRA | 32.10 | 65.00 | 57.14 | 33.06 | 62.50 | 51.34 |
| LoRA (II stage) | 35.45 | 64.29 | **57.74** | **36.86** | 64.29 | 53.29 |
| FFA-LoRA | 33.49 | 62.50 | 50.76 | 32.06 | 65.00 | 53.57 |
| FFA-LoRA (II stage) | 35.88 | 65.38 | 52.24 | 35.53 | 62.50 | 55.38 |
| FedSA-LoRA | 31.13 | 62.50 | 50.20 | 31.58 | 61.90 | 50.26 |
| FedSA-LoRA (II stage) | 35.45 | 63.33 | 51.88 | 35.00 | 64.29 | 52.24 |
| $FedA^2$-LoRA | 36.85 | 64.29 | 54.59 | 33.86 | 65.00 | 54.17 |
| $FedA^2$-LoRA (II stage) | **38.96** | **66.67** | 55.21 | 35.87 | **67.86** | **55.88** |

*Table 9.* Performance comparison across methods for Image and Audio modalities Under the same communication cost.

| Method | Rounds | Image | | | | Audio | | | |
|---|---|---|---|---|---|---|---|---|---|
| | | Acc | $P_{BERT}$ | $R_{BERT}$ | $F_{BERT}$ | Acc | $P_{BERT}$ | $R_{BERT}$ | $F_{BERT}$ |
| LoRA | 33 | 75.00 | 80.14 | 86.12 | 83.03 | 72.22 | 79.88 | **87.46** | 83.46 |
| FFA-LoRA | 66 | **78.57** | 86.74 | **86.85** | **86.63** | 76.92 | 85.18 | 86.54 | 85.72 |
| FedSA-LoRA | 66 | 72.72 | **86.88** | 84.35 | 85.48 | 78.26 | **86.33** | 84.43 | 85.37 |
| FedEx-LoRA | 1 | 61.53 | 79.78 | 80.91 | 80.31 | 55.56 | 79.32 | 84.18 | 81.68 |
| $FedA^2$-LoRA | 33 | 76.92 | 85.58 | 86.16 | 85.93 | **78.84** | 84.94 | 86.97 | **85.94** |

demonstrate that $FedA^2 - LoRA$ serves as a state-of-the-art federated LoRA approach for multimodal settings, effectively handling heterogeneous modality clients without sacrificing performance.

### E.6. ablation study on using Tikhonov Regularization

Without regularization, the solved matrix $B$ can become ill-conditioned, which may accumulate numerical errors across communication rounds and eventually degrade model performance. This effect is clearly reflected in Table 12, $FedA^2 - LoRA(\lambda = 0, w/oTik.)$ only yields marginal gains over vanilla LoRA and even harms some Image metrics (e.g., $F_{BERT}$ drops from 84.11 to 80.96), indicating that the unregularized solution is unstable. In contrast, $FedA^2 - LoRA$ with Tikhonov regularization ($\lambda = 0.1$) achieves a substantial improvement, boosting Image Acc from 64.89 to 76.19 and Audio $F_{BERT}$ from 84.97 to 86.31. These results corroborate the ridge-regression view of Eq.13: a mild Tikhonov term stabilizes the inversion of $AA^T$, leading to more reliable global aggregation. Consistently, the trend observed here on SST2 matches Figure 3(a) in the main text, where a moderate $\lambda$ yields the best overall performance.

### E.7. Evaluation of alternative modality encoders and a different LLM

The results demonstrate that **UniFLoW** is model-agnostic and can be seamlessly applied to different multimodal encoder–LLM combinations.

To further assess the generality and modularity of UniFLoW, we evaluate the framework using *different encoder–LLM pairs*, replacing ImageBind with UniBind(Lyu et al., 2024) and substituting Vicuna-7B with LLaMA-1B(Zhao et al., 2024). As shown in Table 13, UniFLoW consistently improves performance across both image and audio tasks under these alternative backbones, confirming that the proposed analytical aggregation and two-stage training strategy are model-agnostic. Notably, $FedA^2 - LoRA$ and its II stage variant achieve the best overall results, demonstrating that UniFLoW remains effective even when the encoder and LLM architectures differ substantially from the default configuration. This highlights the flexibility and robustness of our framework in real-world multimodal federated settings.

*Table 10.* Performance comparison of zero-padding variants on Image and Audio modalities.

| Method | Image | | | | Audio | | | |
|---|---|---|---|---|---|---|---|---|
| | Acc | $P_{\text{BERT}}$ | $R_{\text{BERT}}$ | $F_{\text{BERT}}$ | Acc | $P_{\text{BERT}}$ | $R_{\text{BERT}}$ | $F_{\text{BERT}}$ |
| – | 61.23 | 78.59 | 81.74 | 80.09 | 50.19 | 78.96 | 81.71 | 80.02 |
| LoRA (zero-padding) | 61.64 | 84.44 | 80.38 | 82.34 | 56.27 | 84.18 | 80.37 | 82.22 |
| **FedA$^2$-LoRA (zero-padding)** | **61.92** | **86.87** | **82.21** | **84.46** | **57.93** | **86.94** | **80.26** | **83.46** |

*Table 11.* Performance comparison across methods on Image and Audio modalities.

| Method | Image | | | | Audio | | | |
|---|---|---|---|---|---|---|---|---|
| | Acc | $P_{\text{BERT}}$ | $R_{\text{BERT}}$ | $F_{\text{BERT}}$ | Acc | $P_{\text{BERT}}$ | $R_{\text{BERT}}$ | $F_{\text{BERT}}$ |
| – | 61.23 | 78.59 | 81.74 | 80.09 | 50.19 | 78.96 | 81.71 | 80.02 |
| LoRA | 64.89 | 86.13 | 82.22 | 84.11 | 58.33 | 82.42 | 87.75 | 84.97 |
| FFA-LoRA | 63.15 | 85.18 | 79.27 | 82.08 | 57.87 | 91.20 | 77.04 | 83.26 |
| FedSA-LoRA | 66.29 | 84.22 | 83.15 | 84.22 | 58.33 | 82.32 | 78.93 | 83.67 |
| FedEx-LoRA | 66.67 | 81.21 | 85.81 | 83.45 | 58.33 | 83.92 | 83.82 | 83.79 |
| $FedA^2 - LoRA$ | **75.00** | 84.51 | 84.34 | 84.43 | 60.17 | 84.97 | 85.83 | **85.39** |

## E.8. Effectiveness of the II Stage Training Strategy

To further understand the role of the proposed II stage training strategy, we compare three scheduling variants: (i) simultaneous training of the encoder and LLM LoRA ($FedA^2 - LoRA$), (ii) a *reverse* schedule that first updates the LLM and then fine-tunes the encoder $FedA^2 - LoRA(Reverse)$, and (iii) our II stage strategy, which first calibrates the modality encoders and then updates the LLM. Intuitively, aligning the modality space in the first stage and only then letting the LLM learn semantics on top of these stabilized representations should be more favorable; reversing this order breaks this dependency and exposes the LLM to highly inconsistent, modality-biased features. The results in Table14 confirm this intuition: while the reverse variant brings only marginal improvements over the vanilla $FedA^2 - LoRA$ baseline, our II stage strategy consistently achieves the best performance on both image and audio metrics (e.g., Image Acc 76.19 and Audio $F_{\text{BERT}}$ 86.31), demonstrating that encoder-first then LLM-second is a more effective training schedule in heterogeneous multimodal FL.

## E.9. Impact of Training Order on Multimodal Federated Optimization

In heterogeneous multimodal FL, different clients hold different modalities (e.g., image, audio, and text), which induces highly inconsistent hidden representations across clients. If we update the encoders and the LLM LoRA simultaneously, these modality-specific representations drive the LLM in conflicting directions, causing it to overfit modality-specific noise and harming both convergence and generalization. To disentangle modality calibration and semantic modeling, we adopt a II stage training schedule: Stage I first fine-tunes the encoders to calibrate the modality space across clients, and Stage II then updates the LLM on top of these stabilized representations. We further compare this design with a coarse-grained schedule that trains the encoder for the first $T = 0.5CR$ communication rounds($CR$)and the LLM for the remaining rounds. As shown in Table 15 this $T$-based variant $FedA^2 - LoRA(T = 0.5CR)$ yields only limited improvements and is consistently worse than our proposed II-stage strategy, whereas $FedA^2 - LoRA$ (II stage) achieves the best performance on both image and audio metrics (e.g., Image Acc 76.19 and Audio $F_{\text{BERT}}$ 86.31). These results confirm that explicitly separating encoder alignment and LLM adaptation at the stage level is more effective than joint or coarse-grained training in heterogeneous multimodal federated settings.

## E.10. Convergence Plot for $FedA^2$-$LoRA$

We further investigate the convergence behavior of $FedA^2 - LoRA$ from two complementary perspectives. First, we report a controlled comparison under the same random seed and learning rate to examine when each method escapes the oscillatory training stage. Second, we visualize the convergence curves of $FedA^2 - LoRA$ using the fine-tuned learning rates corresponding to the main results in Table 4.

*Table 12.* Ablation study on Tikhonov regularization for recovering matrix $B$.

| Method | Image | | | | Audio | | | |
|---|---|---|---|---|---|---|---|---|
| | Acc | $P_{\text{BERT}}$ | $R_{\text{BERT}}$ | $F_{\text{BERT}}$ | Acc | $P_{\text{BERT}}$ | $R_{\text{BERT}}$ | $F_{\text{BERT}}$ |
| LoRA | 64.89 | 86.13 | 82.22 | 84.11 | 58.33 | 82.42 | 87.75 | 84.97 |
| **FedA$^2$-LoRA ( $\lambda=0$, w/o Tik.)** | 65.09 | 83.09 | 78.93 | 80.96 | 62.64 | 86.53 | 82.16 | 84.29 |
| **FedA$^2$-LoRA (Tikhonov, $\lambda=0.1$)** | **76.19** | 83.44 | **88.08** | 85.69 | 60.90 | **87.15** | 85.49 | **86.31** |

*Table 13.* Generalization of UniFLoW with alternative encoders (UniBind) and LLMs (LLaMA-1B).

| Method | Image | | | | Audio | | | |
|---|---|---|---|---|---|---|---|---|
| | Acc | $P_{\text{BERT}}$ | $R_{\text{BERT}}$ | $F_{\text{BERT}}$ | Acc | $P_{\text{BERT}}$ | $R_{\text{BERT}}$ | $F_{\text{BERT}}$ |
| – | 52.17 | 80.85 | 82.47 | 81.61 | 52.94 | 81.92 | 79.36 | 80.61 |
| LoRA | 63.15 | 83.80 | **83.55** | 83.67 | 61.90 | 85.16 | 77.49 | 81.14 |
| FedA$^2$-LoRA | 65.00 | **88.41** | 81.06 | **84.58** | 62.50 | 84.29 | 82.57 | 83.41 |
| **FedA$^2$-LoRA (II stage)** | **66.67** | 83.77 | 83.39 | 83.58 | **63.63** | **86.01** | **84.01** | **85.00** |

The table 16 is designed as a controlled stability analysis rather than a comparison of the best achievable performance. Specifically, all methods are evaluated under the same seed and learning rate, so the results reflect their relative ability to stabilize optimization under identical training conditions. Under this setting, methods that remain trapped in severe oscillation require more communication rounds before entering a stable accuracy region, while methods that escape oscillation earlier exhibit smoother and more reliable optimization. The results show that $FedA^2 - LoRA$ can leave the unstable phase earlier than the baselines, suggesting that the adaptive aggregation strategy helps mitigate unstable local updates and accelerates the transition toward stable convergence.

The convergence plots provide a more detailed view of $FedA^2 - LoRA$ under the learning rates fine-tuned for the main experiments in Table 4. As shown in Figure 5, FedA2-LoRA may still present short-term fluctuations in the initial training rounds, but it rapidly moves out of the oscillatory region and reaches a stable performance plateau. This trend indicates that, after learning-rate tuning, $FedA^2 - LoRA$ not only achieves competitive final performance but also maintains a clear and stable convergence trajectory across different tasks.

Overall, the controlled table and the convergence plots serve different purposes. The table isolates the effect of optimization stability under the same seed and learning rate, highlighting when each method escapes oscillation. The plots, instead, illustrate the practical convergence behavior of $FedA^2 - LoRA$ under the tuned learning rates used for the main Table 4 results.

### E.11. Case Study of $FedA^2$-$LoRA$

To further illustrate the effectiveness of $FedA^2$-LoRA, we conduct case studies on both audio-based and image-based multimodal question answering examples. Different from the aggregate results reported in the main experiments, the case studies provide a more fine-grained comparison of response quality, semantic consistency, and factual alignment with the reference answers.

Table 17 presents an audio-based multimodal QA example. The question asks about the elevation of Mexico City, with the reference answer being 7,350 feet above sea level. Compared with the baseline LoRA model, $FedA^2$-LoRA produces an answer of 7,384 feet, which is closer to the reference than the LoRA prediction of 7,400 feet. This improvement is also reflected in the automatic evaluation metrics, where $FedA^2$-LoRA achieves the highest BERTScore-F1, GPTScore, and BLEURT scores. These results suggest that $FedA^2$-LoRA can better preserve task-relevant semantic information and generate more accurate factual responses.

Table 18 shows an image-based multimodal QA example. The question asks why people on horseback may be present near giraffes. The reference answer explains that the riders could be participating in a guided wildlife tour, ecotourism activity, or conservation-related experience. While the LoRA baseline provides a plausible answer, it mainly focuses on tourists observing giraffes and taking pictures. In contrast, $FedA^2$-LoRA generates a more comprehensive response that

*Table 14.* Comparison of different training strategies for FedA$^2$-LoRA.

| Method | Image | | | | Audio | | | |
|---|---|---|---|---|---|---|---|---|
| | Acc | $P_{\text{BERT}}$ | $R_{\text{BERT}}$ | $F_{\text{BERT}}$ | Acc | $P_{\text{BERT}}$ | $R_{\text{BERT}}$ | $F_{\text{BERT}}$ |
| – | 61.23 | 78.59 | 81.74 | 80.09 | 50.19 | 78.96 | 81.71 | 80.02 |
| LoRA | 64.89 | 86.13 | 82.22 | 84.11 | 58.33 | 82.42 | **87.75** | 84.97 |
| FedA$^2$-LoRA | 75.00 | 84.51 | 84.34 | 84.43 | 60.17 | 84.97 | 85.83 | 85.39 |
| FedA$^2$-LoRA (Reverse) | 69.63 | **86.71** | 81.09 | 83.81 | 60.61 | **87.33** | 82.94 | 85.07 |
| **FedA$^2$-LoRA (II stage)** | **76.19** | 83.44 | **88.08** | **85.69** | **60.90** | 87.15 | 85.49 | **86.31** |

*Table 15.* Effect of different training schedules: simultaneous training, encoder-first (T = 0.5CR), and our II stage strategy.

| Method | Image | | | | Audio | | | |
|---|---|---|---|---|---|---|---|---|
| | Acc | $P_{\text{BERT}}$ | $R_{\text{BERT}}$ | $F_{\text{BERT}}$ | Acc | $P_{\text{BERT}}$ | $R_{\text{BERT}}$ | $F_{\text{BERT}}$ |
| – | 61.23 | 78.59 | 81.74 | 80.09 | 50.19 | 78.96 | 81.71 | 80.02 |
| LoRA | 64.89 | **86.13** | 82.22 | 84.11 | 58.33 | 82.42 | **87.75** | 84.97 |
| FedA$^2$-LoRA | 75.00 | 84.51 | 84.34 | 84.43 | 60.17 | 84.97 | 85.83 | 85.39 |
| FedA$^2$-LoRA (T = 0.5 CR) | 66.15 | 84.03 | 85.78 | 84.90 | 60.35 | **88.98** | 82.72 | 85.73 |
| **FedA$^2$-LoRA (II stage)** | **76.19** | 83.44 | **88.08** | **85.69** | **60.90** | 87.15 | 85.49 | **86.31** |

mentions wildlife tourism, educational field trips, guided rides, and conservation-related activities. This answer is better aligned with the reference and captures a broader set of plausible visual and contextual cues. Consistently, $FedA^2$-LoRA obtains the best scores across BERTScore-F1, GPTScore, and BLEURT.

Overall, these case studies demonstrate that $FedA^2$-LoRA improves not only quantitative performance but also the quality of generated responses in multimodal reasoning scenarios. The model produces answers that are more factually accurate, semantically complete, and better aligned with reference explanations across both audio-based and image-based QA tasks.

## F. Use of Large Language Models (LLMs)

In the preparation of this manuscript, Large Language Models (LLMs) are employed as a general-purpose assistive tool aimed at enhancing the quality, clarity, and presentation of the writing. While the LLMs provide valuable support in specific areas, the core research, experimental design, data analysis, and intellectual contributions remain entirely the work of the authors.

The specific applications of LLMs in this work include:

- **Text Polishing and Refinement**: The LLM is used to review the manuscript for grammatical accuracy, enhance sentence structure, and ensure consistency in phrasing and tone throughout the paper. This process is similar to employing an advanced grammar and style checker, aimed at improving the readability, fluency, and overall quality of the manuscript. The model aids in refining language, ensuring it meets academic writing standards, while maintaining the integrity and originality of the authors' ideas.

- **Coherence and Logical Flow**: We use the LLM to help organize and structure our arguments more effectively. By presenting drafts of sections to the model, we receive suggestions on improving the logical transitions between paragraphs, identifying gaps in the narrative, and strengthening the overall flow. This assistance helps ensure that the document presents a coherent, well-structured, and compelling argument that enhances the readability for our audience.

- **Supplementing and Articulating Ideas**: At various stages of the manuscript preparation, the LLM serves as a sounding board to supplement and articulate the authors' ideas. It assists in expressing complex thoughts more clearly, offering alternative ways to present concepts that are already formulated by the authors. The LLM does not contribute to the generation of new ideas or novel research findings but instead supports the authors in refining their expression, ensuring that their original insights are communicated effectively.

*Table 16.* Performance of different methods on the GLUE benchmark. For all tasks, we report accuracy evaluated across 3 runs with mean and standard deviation.

| | Method | MNLI | RTE | SST2 | QQP | Avg. |
|---|---|---|---|---|---|---|
| LoRA | LoRA | $53.64_{\pm 0.14}$ | $58.13_{\pm 0.17}$ | $54.36_{\pm 0.08}$ | $51.23_{\pm 0.07}$ | 54.28 |
| | FFA-LoRA | $53.65_{\pm 0.16}$ | $58.51_{\pm 0.08}$ | $65.23_{\pm 0.05}$ | $60.40_{\pm 0.12}$ | 59.44 |
| | FedDPA-LoRA | $41.16_{\pm 0.11}$ | $52.70_{\pm 0.22}$ | $50.92_{\pm 0.06}$ | $47.17_{\pm 0.15}$ | 47.98 |
| | FedSA-LoRA | $53.65_{\pm 0.32}$ | $52.84_{\pm 0.05}$ | $53.58_{\pm 0.19}$ | $63.18_{\pm 0.05}$ | 55.81 |
| | $FedA^2$-LoRA | $53.66_{\pm 0.32}$ | $58.56_{\pm 0.22}$ | $69.52_{\pm 0.25}$ | $63.89_{\pm 0.16}$ | 61.43 |
| rsLoRA | rsLoRA | $53.64_{\pm 0.22}$ | $55.90_{\pm 0.03}$ | $69.17_{\pm 0.02}$ | $46.69_{\pm 0.18}$ | 56.35 |
| | FFA-rsLoRA | $52.09_{\pm 0.12}$ | $52.42_{\pm 0.17}$ | $53.97_{\pm 0.15}$ | $52.20_{\pm 0.32}$ | 52.67 |
| | FedDPA-rsLoRA | $53.64_{\pm 0.18}$ | $52.70_{\pm 0.12}$ | $51.95_{\pm 0.21}$ | $54.25_{\pm 0.17}$ | 53.13 |
| | FedSA-rsLoRA | $53.64_{\pm 0.28}$ | $58.47_{\pm 0.17}$ | $60.19_{\pm 0.28}$ | $48.94_{\pm 0.00}$ | 55.31 |
| | $FedA^2$-$rsLoRA$ | $53.66_{\pm 0.10}$ | $59.59_{\pm 0.33}$ | $70.33_{\pm 0.19}$ | $53.30_{\pm 0.28}$ | 59.22 |
| VeRA | VeRA | $53.64_{\pm 0.34}$ | $52.41_{\pm 0.32}$ | $53.58_{\pm 0.21}$ | $59.38_{\pm 0.11}$ | 54.75 |
| | FFA-VeRA | $53.66_{\pm 0.21}$ | $52.05_{\pm 0.10}$ | $53.64_{\pm 0.05}$ | $63.18_{\pm 0.20}$ | 55.63 |
| | FedDPA-VeRA | $53.44_{\pm 0.09}$ | $52.70_{\pm 0.12}$ | $51.94_{\pm 0.11}$ | $53.36_{\pm 0.14}$ | 52.86 |
| | FedSA-VeRA | $53.64_{\pm 0.21}$ | $52.05_{\pm 0.13}$ | $53.84_{\pm 0.12}$ | $56.65_{\pm 0.24}$ | 54.04 |
| | $FedA^2$-$VeLoRA$ | $53.66_{\pm 0.25}$ | $53.46_{\pm 0.00}$ | $54.60_{\pm 0.30}$ | $61.99_{\pm 0.06}$ | 55.92 |

*Table 17.* Quantitative and Qualitative Comparison of Different LoRA Methods on an Audio-Based Multimodal QA Example.

| | - - | LoRA | $FedA^2$-LoRA |
|---|---|---|---|
| BERTScore_F1 | 78.75 | 79.08 | **81.33** |
| GPTScore | 85.00 | 90.00 | **95.00** |
| BLEURT | 23.39 | 25.76 | **28.12** |
| Question: | How man feet above sea level is Mexico City? | | |
| Reference: | 7,350 | | |
| Sample: | 193_0_0.wav | | |
| Output: | Mexico City is located at an elevation of approximately **7,200** feet (2,190 meters) above sea level. | Mexico City is located at an elevation of about **7,400** feet or 2,255 meters above sea level. | Mexico City is approximately **7,384** feet above sea level. |

All suggestions and modifications proposed by the LLM are thoroughly reviewed, edited, and approved by the authors to ensure they accurately reflect the research's intent and underlying meaning. The final responsibility for the content, interpretation, and presentation of this paper rests solely with the authors.

# G. More related work

As privacy concerns grow, companies are increasingly reluctant to upload sensitive data to the cloud for centralized model training. This issue becomes more pronounced in the era of large language models (LLMs) and multi-modal learning, where private data spans diverse modalities such as text, speech, and images. To effectively utilize these distributed data, FedAvg (McMahan et al., 2017) pioneered the federated learning (FL) community and laid the foundation for Parallel Federated Learning (PFL) (Liu et al., 2022). Building upon this paradigm, most federated optimization algorithms improve FedAvg through various enhancements (Li et al., 2020; Duan et al., 2020; Karimireddy et al., 2020; Acar et al., 2021; Qu et al., 2022). For instance, SCAFFOLD (Karimireddy et al., 2020) mitigates client drift through control variates, while FedSAM (Qu et al., 2022) enhances model generalization under heterogeneous client distributions using the Sharpness Aware Minimization optimizer.

Although these methods substantially advance the handling of traditional non-IID issues, they remain limited when applied

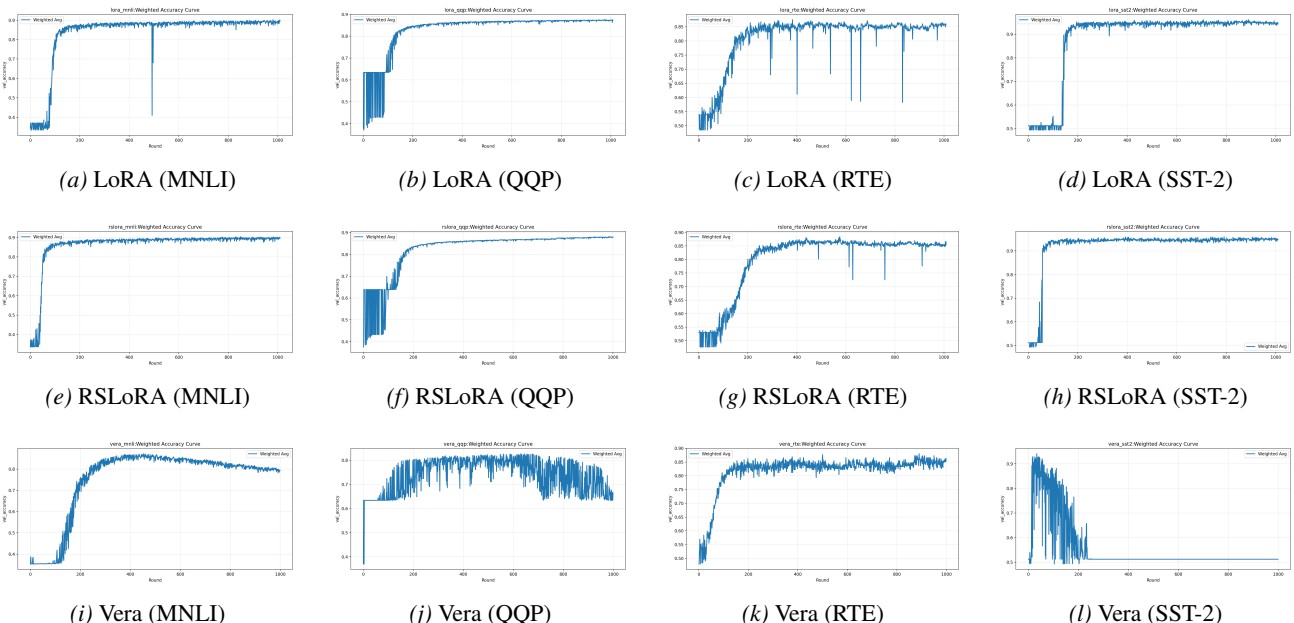

*Figure 5.* Comparison of validation accuracy curves across different methods (Vera, RSLoRA, and LoRA) on four GLUE tasks (MNLI, QQP, RTE, SST-2).

to complex tasks involving large models and multimodal data. Specifically, challenges such as domain shift (Huang et al., 2023) and category shift (Zec et al., 2025) become more severe as model scales and modality heterogeneity increase. To address domain shift, FedCSA (Wang et al., 2023) employs model bias-based clustering to improve global model consistency, while FedDisco (Ye et al., 2023) mitigates poor convergence under category shift. However, these PFL approaches still fall short in fully exploiting client data in large-model scenarios, since each client model is restricted to its local modality-specific data, limiting cross-modal knowledge sharing and undermining the potential of LLM-based multi-modal learning.

## H. Dataset detail

$BERTScore$ (Devlin et al., 2019) is a metric designed to assess the quality of text generation by comparing the contextual embeddings of predicted and reference tokens using the BERT model. It calculates three components: precision (P), recall (R), and F1 score (F1), all based on the cosine similarity between the token embeddings of the predicted and reference sentences. Precision measures how similar the predicted tokens are to the reference tokens, while recall assesses how well the reference tokens are captured by the prediction. The F1 score provides a balanced measure by combining both precision and recall. By using embeddings that capture deeper semantic meaning, $BERTScore$ is particularly useful for tasks like text generation, machine translation, and summarization, where understanding the meaning behind words is more important than the exact word matches. This metric has been shown to align better with human judgment in evaluating the quality of generated text, making it a robust alternative to traditional word-overlap based metrics.

Token accuracy (Acc) (Jiang et al., 2021) is a straightforward metric used to evaluate the performance of a model at the token level. It measures the percentage of tokens (words or subwords) in the model's predicted output that match the corresponding tokens in the reference or ground truth. This metric is particularly useful for tasks such as machine translation, text generation, and token-level classification, where precision in individual token prediction is crucial. Unlike other metrics like sentence-level accuracy, token accuracy provides a more granular view of model performance, allowing for better insight into how well a model captures the structure and meaning of language at the token level.

$HeySQuAD$ (Wu et al., 2024b) is to evaluate a model's ability to understand noisy spoken queries and provide accurate responses. By including both human-spoken and machine-generated questions, it helps assess how models handle variations in spoken language, making it a valuable resource for improving SQA systems. This dataset is particularly useful for training models that deal with real-world conversational scenarios, improving their robustness in noisy environments.

*Table 18.* Quantitative and Qualitative Comparison of Different LoRA Methods on an Image-Based Multimodal QA Example.

| | - - | LoRA | $FedA^2$-LoRA |
|---|---|---|---|
| BERTScore_F1 | 88.44 | 88.86 | **89.34** |
| GPTScore | 33.33 | 75.00 | **85.00** |
| BLEURT | 46.22 | 48.86 | **50.95** |
| Question: | What could be a possible reason for people on horseback to be present near the giraffes? | | |
| Reference: | One possible reason for the people on horseback being near the giraffes could be that they are on a guided wildlife tour, experiencing a thrilling observation of these gentle giants in their natural habitat. This form of eco-friendly tourism allows travelers to connect with nature and appreciate the beauty of wildlife up-close while minimizing their impact on the environment. Additionally, being on horseback can offer a unique perspective and blend more harmoniously into the surroundings, which might lead to more authentic and less intrusive interactions with the animals in their natural environment. | | |
| Sample: | 000000526713.jpg | | |
| Output: | It is likely that the people on horseback near the giraffes are tourists who have come to the zoo to observe and take pictures of the giraffes. | One possible reason for people on horseback to be present near the giraffes is that they are tourists visiting a wildlife reserve or a park where giraffes and other wildlife species are commonly found. These visitors may have come to observe and interact with the giraffes, such as taking pictures or capturing video footage for documentaries or educational purposes. Another possibility is that the people on horseback are park rangers or conservationists, who are monitoring the giraffes' behavior and ensuring their safety in the wild. | A possible reason for people on horseback to be present near the giraffes is that they might be involved in a wildlife tourism activity or an educational field trip. In the image, a group of people is seen mounted on horses, possibly exploring the African savannah alongside the giraffes. This scenario is not uncommon in many African countries, where tourists come to observe, learn about, and engage in activities related to wildlife conservation and the environment. |

$PandaGPT's$ (Su et al., 2023) visual instruction dataset is designed to improve multimodal instruction-following models by providing a collection of image-language instruction pairs. This dataset enables models to learn how to process and respond to multimodal inputs, combining images with textual instructions and responses. It plays a crucial role in training models like PandaGPT to handle complex tasks across different modalities, enhancing their ability to generate responses based on visual and textual inputs.

