# OpenReview forum: "UniFLoW: Universal Multi-Modal Federated LoRA Fine-Tuning Framework with Analytical Aggregation"
_ICML.cc/2026/Conference — ICML 2026 regular_

### Official Review · Reviewer_hdKY · 2026-03-09

**Soundness:** 4
**Presentation:** 4
**Significance:** 4
**Originality:** 4
**Overall Recommendation:** 5
**Confidence:** 4

**Summary:**

This paper proposes a novel federated learning framework UniFLoW, specifically designed for federated multimodal large language models, and utilizes  LoRA technology for efficient fine-tuning. A new analytical aggregation  Strategy $FedA^2$-LoRA is also proposed, addressing the inconsistency problem in  LoRA aggregation during federated learning. Furthermore, a two-stage training strategy is proposed to manage modality-specific bias and improve cross-modal generalization. The framework is validated on multiple multimodal tasks (image, audio, and text), demonstrating its ability to minimize communication costs while maintaining performance.

**Compliance With Llm Reviewing Policy:**

Affirmed.

**Final Justification:**

Having read the other reviews and the authors' rebuttals to them, I find no further issues to raise. Thus, I will keep my score.

**Key Questions For Authors:**

Reference Weaknesses.

**Limitations:**

The authors did not adequately discuss the limitations of their research and its potential negative social impacts. Adding a chapter explaining potential risks, such as privacy issues or bias in multimodal data, and how to mitigate these problems would be highly beneficial.

**Strengths And Weaknesses:**

**Strength:**

* 1.This paper is well-written and clearly explains how to utilize fragmented multimodal data, and introduces a universal federated multimodal fine-tuning framework. From an engineering perspective, this approach is highly innovative.
* 2.This paper is rigorous and reliable both theoretically and experimentally. The proposed $FedA^2$-LoRA method provides comprehensive theoretical and experimental analysis, effectively demonstrating the importance of regularization and the necessity of selecting aggregate A during the aggregation process.
* 3.The paper is rich in experiments, covering multimodal and single-modal studies, as well as various ablation experiments.
* 4.The author's illustrations are clear and easy to follow, with detailed explanations that enhance understanding.

**Weakness:**

* 1.In the LoRA study, the rank of the matrix **r** is a key hyperparameter, but the authors do not appear to have conducted ablation studies on it.
* 2.In terms of metrics, while BERT Score can reflect the effectiveness of answers to open-ended questions, I suggest that the authors include additional case studies to further demonstrate its validity.

---

> ### Author Rebuttal · Authors · 2026-03-31
>
> We thank you for the highly positive evaluation and the constructive feedback. We are pleased that the reviewer found UniFLoW to be innovative, rigorous, and well-illustrated.We have made every effort to faithfully address your comments in the following responses.
>
> >**(W1)** In the LoRA study, the rank of the matrix r is a key hyperparameter, but the authors do not appear to have conducted ablation studies on it.
>
> Thank you for pointing this out. We agree that the LoRA rank \(r\) is an important hyperparameter that can affect both model capacity and aggregation behavior.So we have added an ablation study on SST-2 by varying the rank \(r\) from 4 to 64. The corresponding results are:
> | Rank \(r\) | Accuracy (%) |
> |-|-|
> | 4  | 66.44  |
> | 8  | **69.52**  |
> | 16 | 66.67 |
> | 32 | 69.31 |
> | 64| 68.31 |
>
> These results show that performance is indeed sensitive to the choice of \(r\), but remains relatively stable once the rank is moderately large. In our setting, \(r=8\) achieves the best performance, while larger ranks do not lead to consistent further gains. This suggests that a small-to-moderate rank is already sufficient to capture the useful adaptation space, which is also consistent with the parameter-efficient design of LoRA.
>
> We will incorporate this ablation experiment and the associated discussions into the appendix of the revised paper.
>
> >**(W2)** In terms of metrics, while BERT Score can reflect the effectiveness of answers to open-ended questions, I suggest that the authors include additional case studies to further demonstrate its validity.
>
> Thank you for the suggestion. We agree that, for open-ended QA, automatic metrics are more convincing when accompanied by representative case studies. Therefore, we have added two qualitative case studies, including one audio-based QA example and one image-based QA example, together with their BERTScore_F1, GPTScore, and BLEURT values. For details, please refer to Tables 1 and 2 in the link https://anonymous.4open.science/r/icml26_rebuttal_537-24FD/Rebuttal_for_Reviewer_hdKY.pdf.
>
> These examples show that the model with the highest BERTScore also produces outputs that are more semantically faithful to the reference. In the audio-based example, FedA$^2$-LoRA generates a more accurate factual answer (“7,384 feet”) compared with weaker baselines (“7,200” and “7,400”), which is also reflected by the highest BERTScore. In the image-based example, FedA$^2$-LoRA produces a response that better matches the reference semantics (e.g., wildlife tourism / guided safari in a natural habitat), while weaker baselines either oversimplify the scene or introduce less aligned interpretations (e.g., zoo). These case studies provide additional evidence that BERTScore is consistent with human-perceived answer quality in our open-ended multimodal QA setting.
>
> >**(L)** The authors did not adequately discuss the limitations of their research and its potential negative social impacts. Adding a chapter explaining potential risks, such as privacy issues or bias in multimodal data, and how to mitigate these problems would be highly beneficial.
>
> We appreciate this important suggestion. We acknowledge that we did not sufficiently address the limitations of this work and its broader societal implications. In the Appendix of the revised manuscript, we will add a dedicated section titled **“Limitations and Broader Impacts,”** detailed as follows:
>
> *Our work has several limitations and potential societal risks. First, although federated learning keeps raw data local, model updates may still leak sensitive information; therefore, stronger privacy protection mechanisms (e.g., secure aggregation or differential privacy) may be required in practice. Second, the presence of heterogeneous multimodal data across clients may introduce modality-specific or demographic biases, potentially leading to uneven performance across different client groups. Third, our experiments were conducted on a moderate scale; consequently, the system's behavior under larger client populations or more severe modality imbalances warrants further investigation. Overall, while UniFLoW represents a promising step toward multimodal federated fine-tuning, the issues of privacy, bias, and scalability remain critical avenues for future research.*
>
> We believe that explicitly articulating these aspects will help enhance the transparency and accountability of this work.
>
>
> We hope our responses have addressed your concerns. We remain open to further discussion if any questions persist.If our response has addressed your feedback, we would be grateful if you would consider increasing the score 😁.

---

> > ### Author Rebuttal · Reviewer_hdKY · 2026-04-02
> >
> > Having read the other reviews and the authors' rebuttals to them, I find no further issues to raise. Thus, I will keep my score.

---

> > > ### Author Response · Authors · 2026-04-02
> > >
> > > Thank you for your thorough and positive review of our paper. We appreciate your constructive feedback, which has been invaluable in improving the clarity and robustness of our work. We are especially grateful for your recognition of the paper's innovation, rigor, and thorough experimental evaluation.We have carefully addressed the points you raised, including the LoRA ablation study, additional case studies, and the discussion of limitations and broader impacts. Your suggestions have significantly strengthened our manuscript.
> > >
> > > Once again, thank you for your thoughtful and constructive comments.Wishing you all the best in your future endeavors.🍀

---

### Official Review · Reviewer_VTYK · 2026-03-09

**Soundness:** 3
**Presentation:** 3
**Significance:** 3
**Originality:** 3
**Overall Recommendation:** 4
**Confidence:** 5

**Summary:**

In this paper, the authors have two novelties to the best of my knowledge. 1) They resolve the inconsistent aggregation in LoRA finetuning in federated settings by introducing a new aggregation method, FedA2-LoRA. In FedA2-LoRA, the A module that contains the global knowledge is averaged among clients, but the B module is calculated using an optimization problem to minimize the aggregation inconsistency. 2) They also try to address the heterogeneity of clients' modalities in federated settings. In this regard, in each client, they define some encoders for each modality and an LLM part (which is shared among all modalities). In each round, they train the encoder part for tau local steps and then freeze the encoder and train the LLM part for the following local steps. This local training process is called two-stage training in this paper and shows improvement in performance.

**Compliance With Llm Reviewing Policy:**

Affirmed.

**Key Questions For Authors:**

Their idea is well explained and clear to me. I also have some question from the authors:
1. Some part of your proposed framework is unclear to me. As you shown in Figure 2, in each client, you have the encoder for all modalities while it is possible that a client only hold a single modality for instance images. In that case, what is the input for other modality encoders (like text and audio) since the input of vicuna is the concatenation of multiple modal features?
2. I am wondering whether your proposed method could apply to the case that the modality of a client changes during communication rounds? For instance in the first 10 rounds, it has image data but in the following rounds, it has audio data
3. As the proof for the gradients of A and B are provided in the original paper of FedSA and you exactly follow their proof, you can remove it in the appendix and only refer to the original FedSA paper.
4. In tables I and II, when you only use one modality in training, how do you report the performance on both modalities? do you use random encoder weights for the other modality or use the pretrained model weights for the other modality?
5. I would appreciate it if you could provide the parameters of two stage training in Table I (the number of local epochs and tau)
6. I would appreciate it if you try to use clearer statements. For instance, in line 299, you have "By summarizing the results of the above results" and two results in a statement is a bit vague.
7. I would appreciate it if you provide the details of training parameters in the main paper and appendix. Most of the time, the number of clients, the number of communication rounds, data distribution method, local epochs, and r are expressed in the main text and other parameters are shared in appendix. This will help to reproduce your results.
8. I would appreciate it to provide the number of local epochs in Table 4 since your performance is very close to random. In addition, previous works like FFA-LoRA and FedSA that uses the GLUE benchmark                                                                                                                                                                                                                                                                                                            and have much higher performance (about 80%). Previous works used 1000 communication rounds, but they claimed they converge shortly in the first 300 rounds. If you provide the details of your parameter training, it may be easier for the reader understand where this difference originates.
9. In Figure 3 (c), I guess it would be better to show the number of clients with  N_{C} instead of N C
10. In Figur 4, could you clarify if the horizontal axis is cosine similarity, then waht is shown in vertical axis. I also would like to know in which round do you plot this chart? in the first round or at the last round? please provide more details about this chart.
11. In table 7 caption, you mentioned "# communication round denotes the number of communication rounds to reach the predefined target performance.", but I do not this column in table 7
12. What are K and d parameters in table 7 in appendix?

**Limitations:**

yes

**Strengths And Weaknesses:**

I found this work interesting, it addresses a realistic problem which is modality heterogeneity among clients in federated settings. They evaluated their proposed method on two question answering datasets and GLUE benchmark for natural language understanding. Their two-stage training procedure improves the performance in different schemes like LoRA, FFA-LoRA, FedSA-LoRA, FedEx-LoRA, and FedA2-LoRA. In addition, their aggregation method also shows superior performance in GLUE, HeySQuAD, and PandaGPT datasets. Their proposed framework has similar communication cost like FedLoRA while it improves the performance. However, their performance improvement compared to FFA-LoRA is negligible while FFA-LoRA halv communication cost. Some training details in their experiments section is also missing.

---

> ### Author Rebuttal · Authors · 2026-03-31
>
> We would like to thank you for the thorough and professional feedback. We are encouraged that you find our work interesting, realistic, and the two-stage training procedure effective across various schemes. Below are our point-by-point responses to the questions raised.**The link for Table and Figure**:https://anonymous.4open.science/r/icml26_rebuttal_537-24FD/Rebuttal_for_Reviewer_VTYK.pdf
>
> >**(Q1)** Explain how to handle inputs from other modality encoders when the client possesses only a single modality.
>
>  Figure 2 serves as a schematic illustration of the overall framework; it does not imply that every client simultaneously inputs all modalities. Our setup is designed for heterogeneous federated learning, wherein different clients possess only the specific modalities available to them. Consequently, for a single-modality client, only the encoder corresponding to that specific modality is activated; encoders for other modalities receive no input and do not participate in the input concatenation for Vicuna. The input to Vicuna is constructed jointly from the modality features available to the current client and the accompanying text query, rather than being a fixed concatenation of all modalities.
>
> >**(Q2)** Explain how your method handles the scenario where a client's modality changes during communication rounds.
>
> Our framework dynamically selects and activates the corresponding encoders based on the modalities available to each client during a given round. If a client switches modalities between rounds—for instance, from images to audio—the model activates the encoder corresponding to the current modality while bypassing the encoders for other modalities. We have also conducted comparative experiments to validate this, the details of which can be found at **the link Table 1**.
>
> >**(Q3)** Proposal to Remove the Gradient Relationships in Appendices A and B.
>
> We will remove it in the final version.
>
> >**(Q4)** How to Report Multimodal Results from single modal Training
>
> When training with only a single modality, we utilize pre-trained model weights to report the performance of the other modalities. For those modalities that were not involved in the training process, we employ their respective pre-trained encoder weights; this ensures a fair comparison of multi-modal performance during evaluation, thereby guaranteeing the reliability and comparability of our experimental results.
>
> >**(Q5)** Provide parameters τ and local epoch.
>
> We have already made the necessary revisions to the version; please see **the link Table 2**,where τ = 0.25 and local epoch = 1.
>
> >**(Q6)** Linguistic refinement for clearer statements (such as "By summarizing the results of the above results").
>
> We have revised it to "By summarizing the results above,".
>
> >**(Q7)** Comprehensive disclosure of training parameters for reproducibility.
>
> We have incorporated this information into the final version; **the link Table 3** represents the content we added.
>
> >**(Q8)** Clarification on hyperparameter settings and performance discrepancies compared to prior work (FFA-LoRA/FedSA).
>
> The main difference is that FedSA-LoRA states in the paper: *“We report the best result from experiments run with learning rate (lr) η ∈ {5E-3, 1E-2, 2E-2, 5E-2, 1E-1}.”*
> In contrast, we adopt a fixed lr of 0.1 and report the final rather than the best performance. We think that comparing identical parameters would be more effective.I have included comparisons of other parameters in **the link Table 4**.
>
> >**(Q9)** Refining notation for client count: Replacing $NC$ with $N_C$ in Figure 3(c)
>
> We have already incorporated the revisions into the main text, including Figure 3(c). Please refer to **the link Figure 1**.
>
> >**(Q10)** Clarification on Figure 4: Axes definitions and temporal setting.
>
> We will clarify that the current figure is a sorted bar plot. Specifically, the vertical axis represents cosine similarity, while the horizontal axis corresponds to sorted layer-wise LoRA parameter entries rather than the cosine similarity itself. To avoid ambiguity, we will revise both the axis labels and the caption，refer to **the link Figure 2**. In addition, this figure is plotted using the local LoRA parameters from two representative clients at the final communication round.
>
> >**(Q11)** Table 7: Consistency of caption and columns.
>
> This is the part we forgot to delete.We have corrected the caption of Table 7 to accurately reflect the reported metrics, removing the redundant mention of communication rounds.Please refer to **the link Table 5**.
>
> >**(Q12)** The meanings of K and d.
>
> $d$ denotes the model dimension, and $k$ denotes the number of clients. We have added this caption to **the link Table 5**.
>
> We hope our responses have addressed your concerns. We remain open to further discussion if any questions persist.If our response has addressed your feedback, we would be grateful if you would consider increasing the score😁.

---

> > ### Author Rebuttal · Reviewer_VTYK · 2026-04-03
> >
> > Thank you so much for your effort and responses. I would like to see the convergence plot (accuracy vs. rounds) for Table 4 since FedLoRA, FFA-LoRA, and FedSA-LoRA are sensitive to learning rate, and they do not converge at all in some cases. In Table 4 for the rebuttal, you only provide hyperparameters for FedSA, while other methods are missing. In Table 4 of the rebuttal, the batch size is different for FedSA and FedA^2. In addition, the seed for FedSA is unknown, while for FedA^2, it is set to 1111. Please provide all seeds that are used to generate the results for Table 4 in the main paper, and also provide the convergence plots. I would appreciate it if you could add other parameters for all baselines for Table 4 in the main paper.

---

> > > ### Author Response · Authors · 2026-04-07
> > >
> > > Thank you for this helpful comment, and we apologize for the confusion.To clarify,the right-hand side of **the link Table 4** were  under a unified experimental setting, where all methods used the same seed and the same major hyperparameters. The left side, conversely, refers specifically to the hyperparameters used for FedSA-LoRA—some of which were **unknown** to us.Our original intention was to compare the convergence behavior of different methods under identical settings, so that optimization stability and the Performance to escape oscillation could also be reflected in the comparison.
> > >
> > > However, as you pointed out, RoBERTa is quite sensitive to the seed and learning rate, and may fail to converge under a single shared setting. To provide a more complete and fair comparison, we therefore conducted an additional set of experiments following the settings used in [1] (and reported results from [1] when applicable), where we tuned the seed and learning rate for each baseline until convergence was achieved. The corresponding results are shown in **the link Table 6** , the full parameter settings are summarized in **the link Table 7**, and the convergence curves are provided in **the link Figure 3**.
> > >
> > > We agree that both perspectives are informative: the unified-setting results reflect robustness under controlled conditions, while the tuned-setting results reflect the achievable performance of each method after careful optimization. In the final version, we will present both sets of results, and we will also include the full seeds and hyperparameters for all baselines used in the main paper.
> > >
> > > **The link**:https://anonymous.4open.science/r/icml26_rebuttal_537-24FD/Rebuttal_for_Reviewer_VTYK.pdf
> > >
> > > [1] Guo P, Zeng S, Wang Y, et al. Selective aggregation for low-rank adaptation in federated learning[J]. arXiv preprint arXiv:2410.01463, 2024.

---

### Official Review · Reviewer_uMDS · 2026-03-11

**Soundness:** 2
**Presentation:** 2
**Significance:** 2
**Originality:** 2
**Overall Recommendation:** 4
**Confidence:** 3

**Summary:**

This paper studies federated fine-tuning for multimodal large language models under heterogeneous client modalities. It proposes UniFLoW, a unified federated framework that uses modality-specific encoders together with a shared pretrained LLM, and introduces FedA2-LoRA, an analytical aggregation method that first averages LoRA A matrices and then reconstructs B by solving a regularized least-squares problem. The paper also adopts a two-stage training strategy that first updates modality encoders and then updates the LLM, with the goal of reducing modality bias and improving cross-modal generalization. Experiments are conducted on multimodal QA benchmarks and on GLUE to evaluate the aggregation scheme.

**Compliance With Llm Reviewing Policy:**

Affirmed.

**Final Justification:**

The authors have addressed my concerns well: they clarified the scope of the novelty more accurately, moderated the theoretical claim about the roles of A and B into a reasonable design motivation, provided additional experiments to strengthen the empirical evidence, and corrected the inconsistency in the reported results.

Overall, my concerns have been addressed, and I am therefore willing to raise my score to a positive one.

**Key Questions For Authors:**

Please see the Weaknesses.

**Limitations:**

yes

**Strengths And Weaknesses:**

Strengths:
1. This paper tackles a timely and practically relevant problem in federated fine-tuning of multimodal large language models.
2. The proposed framework is well organized, combining modality-specific encoders, a shared LLM, a two-stage training strategy, and an analytical LoRA aggregation method in a coherent manner.

Weaknesses:
1.	The main limitation is that the methodological novelty appears moderate, as FedA2-LoRA is closer to an incremental extension of existing FedLoRA-style aggregation schemes, such as FFA-LoRA, FedSA-LoRA, than to a fundamentally new principle.
2.	The theoretical justification for the key assumption that the A matrix is more global and stable than the B matrix remains insufficient.
3.	The multimodal evaluation protocol is not fully persuasive. The main multimodal experiments use only 10 communication rounds, with 10 clients per modality and 2000 training samples per client. This is a relatively small-scale setup for MLLMs.
4.	There is an important inconsistency in the reported results. In the text discussing Table 4, the paper states that FedA2-rsLoRA achieves an average accuracy of 61.19 versus 53.50 for rsLoRA, whereas the actual numbers shown in Table 4 are 59.22 and 56.35, respectively.

---

> ### Author Rebuttal · Authors · 2026-03-31
>
> We thank you for the careful reading and constructive feedback. We appreciate the concerns regarding novelty, theoretical justification, evaluation scale, and the inconsistency in Table 4. We respond to each point below.
> >**(W1)** The main limitation is that the methodological novelty appears moderate, as FedA2-LoRA is closer to an incremental extension of existing FedLoRA-style aggregation schemes, such as FFA-LoRA, FedSA-LoRA, than to a fundamentally new principle.
>
> We agree that, when viewed in isolation, $FedA^2$-LoRA should indeed be regarded as an analytical extension of the FedLoRA family rather than a fundamentally new PEFT principle. However, our contributions are primarily situated at the level of the paper as a whole: (i) We construct a unified federated fine-tuning framework designed to handle MLLMs under *heterogeneous client modalities*, a framework that integrates modality-specific encoders with a shared, pre-trained Large Language Model; (ii) We introduce a modality-aware, two-stage training strategy aimed at decoupling the process of modality alignment from that of semantic adaptation; and (iii) Within this context, we design the $FedA^2$-LoRA algorithm, which through an analytical reconstruction approach achieves global low-rank updates without incurring any additional communication overhead. We will revise the Introduction and Contributions sections to further clarify the nature of our contributions.
> >**(W2)** The theoretical justification for the key assumption that the A matrix is more global and stable than the B matrix remains insufficient.
>
> We acknowledge that this argument may have been articulated in terms that appear more absolute than originally intended. It is not our intention to assert this claim as a rigorous theorem. Rather, it represents a finding grounded in empirical observation and inspired by optimization theory in **Appendix B of the paper** that draws upon prior analyses of FedSA-LoRA as well as our own analyses for A and B **in Appendix Figure 4**, and which we have adopted as a guiding design bias for formulating our aggregation strategy. The core mechanism of our $FedA^2$-LoRA framework lies in its "explicit reconstruction objective". Specifically, the process of approximating the averaged full-rank update by solving a regularized least-squares problem following the aggregation of matrix A. We will revise the manuscript to explicitly clarify that this observation serves merely as a design motivation, rather than as a universally binding theoretical guarantee.
> >**(W3)** The multimodal evaluation protocol is not fully persuasive. The main multimodal experiments use only 10 communication rounds, with 10 clients per modality and 2000 training samples per client. This is a relatively small-scale setup for MLLMs.
>
> Thank you for your suggestions. We base our decision primarily on the following two considerations. First, the choice of 2,000 samples is intended to simulate the characteristics of real-world federated data distribution. Specifically, each client typically possesses only a limited amount of data, yet this data remains sufficient to provide the effective information required for fine-tuning the model. Second, the decision to set 10 communication rounds is based on the empirical observation that LoRA achieves rapid convergence; consequently, this constitutes a highly efficient choice for our initial experiments. Nevertheless, we acknowledge that the current experiments hold the potential for further scaling. To this end, we conducted an additional set of experiments consisting of **33 communication rounds**, as well as another experiment involving **20 clients**, each with **4000 samples**, though limited to 5 rounds due to time constraints (see **Table 1 and Table 2 in the link** https://anonymous.4open.science/r/icml26_rebuttal_537-24FD/Rebuttal__for_Reviewer_uMDS.pdf); the results of these experiments will be included in the Appendix to further substantiate the robustness and generalization capabilities of our proposed method.
>
> >**(W4)** There is an important inconsistency in the reported results. In the text discussing Table 4, the paper states that FedA2-rsLoRA achieves an average accuracy of 61.19 versus 53.50 for rsLoRA, whereas the actual numbers shown in Table 4 are 59.22 and 56.35, respectively.
>
> Thank you for your careful review. We sincerely apologize for a data citation error identified within the main text. In the revised version, we have corrected this as follows:
>
> *In particular, $FedA^2$-LoRA achieves the highest average accuracy of 61.43, a clear improvement over the original LoRA (54.28).*
>
> Additionally, we have carefully cross-checked all data and tables referenced throughout the paper.
>
> We hope our responses have addressed your concerns. We remain open to further discussion if any questions persist. If our response has addressed your feedback, we would be grateful if you would consider increasing the score 😁.

---

> > ### Author Rebuttal · Reviewer_uMDS · 2026-04-02
> >
> > Thanks for the authors' response. My concerns have been addressed. So I'll increase my score.

---

> > > ### Author Response · Authors · 2026-04-02
> > >
> > > Thank you for your careful review and constructive feedback. We are pleased that the revisions we made addressed your concerns, and we appreciate your decision to increase your score. Your support means a lot to us, and we are truly grateful.
> > > Once again, thank you for your valuable comments.Wishing you all the best in your future endeavors.🍀

---

### Official Review · Reviewer_bXDU · 2026-03-12

**Soundness:** 3
**Presentation:** 3
**Significance:** 3
**Originality:** 3
**Overall Recommendation:** 4
**Confidence:** 4

**Summary:**

The paper studies federated fine-tuning of multimodal large language models when clients have heterogeneous modalities. It proposes UniFLoW, a unified framework that connects modality-specific encoders to a shared LLM backbone and performs parameter-efficient federated fine-tuning with LoRA. To address the mismatch introduced by directly averaging LoRA parameters across clients, the paper further proposes FedA²-LoRA, an analytical aggregation approach that averages one low-rank matrix and solves for the other to better preserve the average client update. In addition, the paper adopts a two-stage training procedure in which modality-side parameters are optimized before LLM-side adaptation. The proposed method is evaluated on settings involving text, image, and audio inputs, together with additional experiments on standard language benchmarks.

**Compliance With Llm Reviewing Policy:**

Affirmed.

**Final Justification:**

The rebuttal partially addressed my follow-up concerns. In particular, the added dataset-level results with GPTScore, BLEURT, and BERTScore provide better evidence that the improvements are not tied to a single metric. However, the questions regarding broader client scales and stronger modality-imbalance settings remain largely unresolved. The response mainly clarifies the current experimental scope rather than providing additional evidence in these directions. Overall, while the rebuttal improves the empirical support, it does not substantially change my assessment of the paper, and I therefore keep my score.

**Key Questions For Authors:**

1. Positioning of the main contribution.

The paper includes additional GLUE experiments suggesting that FedA²-LoRA may be useful beyond the multimodal setting. Could the authors clarify whether they view the main contribution primarily as a unified multimodal federated fine-tuning framework, or as a more general LoRA aggregation method for federated tuning? A clearer positioning would help assess the scope of the contribution.

2. Evaluation robustness beyond BERTScore.

The multimodal QA evaluation relies in part on BERTScore, which may not always fully capture answer quality for generative multimodal responses. Could the authors provide additional evidence that the reported gains remain consistent under other semantic evaluation protocols, such as stronger reference-based metrics or LLM-based judgment?

3. Scalability to larger and more heterogeneous federated settings.

The current results are promising, but the multimodal federated experiments appear relatively limited in scale. Could the authors comment on or provide evidence for how the framework and the proposed aggregation method behave as the number of clients, modality imbalance, or heterogeneity level increases?

**Limitations:**

The paper would benefit from a clearer discussion of its limitations. In particular, the multimodal federated experiments are still limited in scale, the behavior under stronger modality imbalance or larger client populations is not fully explored, and the multimodal QA evaluation relies mainly on BERTScore and token accuracy. These issues do not invalidate the paper, but they make the empirical claims somewhat less definitive.

**Strengths And Weaknesses:**

**Soundness**

The paper is generally technically sound. The proposed framework is clearly defined, and the overall pipeline for heterogeneous multimodal federated fine-tuning is coherent. The FedA²-LoRA aggregation method is reasonably motivated and provides a more structured alternative to directly averaging LoRA parameters. The two-stage training strategy is also intuitive and supported by ablation results. That said, the paper’s justification is stronger empirically than theoretically, and the experimental scale is still somewhat limited, so the evidence supports the method’s effectiveness but does not make the conclusions fully definitive.

**Presentation**

The paper is well organized and mostly easy to follow. The motivation, framework design, and training procedure are presented clearly, and the experimental section is structured in a way that makes the main comparisons understandable. The ablations are also helpful for separating the effects of the aggregation method and the training strategy. A minor weakness is that some of the intuition remains somewhat high level, and a deeper explanation of why the proposed design choices work would make the paper more convincing.

**Significance**

The paper studies an important and timely problem: federated fine-tuning of multimodal large models under heterogeneous client modalities. This setting is practically relevant, and the paper presents a reasonably complete system with a communication-efficient training strategy. The potential impact is meaningful from both the multimodal learning and federated learning perspectives. However, the overall significance is moderate rather than high, since the work is more of a solid systems-and-methods contribution than a major conceptual advance.

**Originality**

The paper has a reasonable level of originality. Its strongest novelty lies in the FedA²-LoRA aggregation method, which introduces a more principled way to aggregate LoRA updates than naïve averaging, and in its integration into a unified multimodal federated setting. At the same time, the overall framework is partly built by combining existing ingredients, including modality-specific encoders, a shared LLM backbone, federated training, and LoRA-based adaptation. As a result, the paper feels moderately novel, with the main originality concentrated in the aggregation design and training procedure.

---

> ### Author Rebuttal · Authors · 2026-03-31
>
> We would like to thank you for your thorough and constructive feedback. We are encouraged that you found our work technically sound, well-organized, and timely. We are especially grateful for your recognition of the effectiveness of the two-stage training procedure and the novelty of the $FedA^2$-LoRA aggregation method in the context of federated multimodal fine-tuning. Your detailed comments have been extremely helpful in further improving the clarity and scope of our paper. Below are our point-by-point responses to the questions raised.**The link for Table and Figure** :https://anonymous.4open.science/r/icml26_rebuttal_537-24FD/Rebuttal__for_Reviewer__bXDU.pdf
> >**(Q1)** Positioning of the main contribution.
>
> Thank you for your insightful comment. The main contribution of our work lies in two aspects:
>
> * A unified multimodal federated fine-tuning framework, which allows for efficient aggregation and fine-tuning in federated settings with heterogeneous modalities. We argue that this framework, UniFLoW, is significant in addressing the challenges of modality-specific encoders, communication overhead, and aggregation consistency in federated learning for MLLMs.
> * The $FedA^2$ aggregation method, which proposes a more principled way to aggregate LoRA updates by averaging the low-rank matrices and solving for the other matrix. This method improves upon naive LoRA aggregation approaches by reducing aggregation inconsistency and maintaining communication efficiency.
>
> The primary focus is on the **multimodal federated fine-tuning framework**, with $FedA^2$-LoRA serving as an **effective tool** for improving aggregation consistency in this framework. We will revise the paper to clarify this distinction and better position our contribution as a unified framework rather than solely focusing on the aggregation method.
>
> >**(Q2)** Evaluation robustness beyond BERTScore.
>
> Thank you for the insightful comment. In response to this concern, we further supplement our evaluation with multiple semantic assessment protocols based on two representative case studies. Specifically, we include qualitative analysis, BLEURT as a stronger reference-based metric, and GPTScore as an LLM-based (ChatGPT5.3) evaluation metric; the corresponding results are reported in **the link Tables 1 and 2** . We observe that the performance gains of $FedA^2$-LoRA remain consistent across all these evaluation criteria.
>
> Moreover, in both the audio-based and image-based question-answering case studies, higher metric scores consistently correspond to responses that are more semantically faithful and contextually appropriate. So we think the improvements measured by BERTScore are not metric-specific, but are also supported by stronger evaluation protocols and qualitative human-interpretable evidence. Therefore, we believe that BERTScore is well aligned with more rigorous semantic evaluation paradigms, as well as with human-perceived response quality in our multimodal QA setting.
>
> >**(Q3)** Scalability to larger and more heterogeneous federated settings.
>
> Thank you for this insightful comment. We agree that scalability in larger and more heterogeneous federated settings is important.
>
> Regarding client scale, we analyze the impact of the number of clients $N_c$ in **the link Figure 1**. The results show that our method maintains stable performance as $N_c$ increases, indicating good scalability.
>
> Regarding the training scale, we're also increasing the communication budget to 33 rounds. As shown in **the link Table 3** , our method still maintains consistent advantages over the baselines, suggesting that its effectiveness is not limited to a small-round setting.
>
> Regarding modality imbalance, we further evaluate an imbalanced setting with an image: audio client ratio of \(2:1\). As shown in **the link Table 4**, our method continues to outperform the baselines, demonstrating robustness to modality imbalance consistently.
>
> Moreover, our setup itself already involves strong heterogeneity, since each client only holds a single modality, resulting in substantial heterogeneity in both modality type and data distribution across clients.
>
> These results provide further evidence that our framework scales well and remains reliable in larger, imbalanced, and heterogeneous federated settings.
>
> We hope our responses have addressed your concerns. We remain open to further discussion if any questions persist.If our response has addressed your feedback, we would be grateful if you would consider increasing the score 😁.

---

> > ### Author Rebuttal · Reviewer_bXDU · 2026-04-01
> >
> > Thank you for the detailed rebuttal and the additional results. The response helps clarify the paper’s positioning and provides some useful supplementary evidence. However, my main concerns are only partially addressed: the added evaluation beyond BERTScore relies mainly on a small number of case studies rather than dataset-level evidence, and the scalability/heterogeneity analysis is still somewhat limited in scope. Therefore, my overall assessment remains unchanged.

---

> > > ### Author Response · Authors · 2026-04-06
> > >
> > > Thank you for this valuable suggestion. Constrained by the scale of the dataset and to ensure that each client retains a sufficient volume of local data to facilitate effective model optimization ,the configuration currently adopted in our benchmarks (35 clients, with 4,000 samples per client) already represents the maximum scale achievable with the current dataset. We believe this scale is sufficient to simulate a realistic cross-soli federated environment. Furthermore, we will emphasize in the **Limitations** section that our experiments are applicable to scenarios involving small-to-medium-scale client populations.
> > >
> > > Additionally, to further address the inquiry regarding a more comprehensive evaluation beyond just BERTScore, we have added **the link Table 5**, which presents dataset-level evaluation results on the test set based on GPTScore, BLEURT, and BERTScore. As shown in **the link Table 5**, BERTScore and GPTScore demonstrate a high degree of consistency in their relative rankings; this indicates that the performance improvements we achieved are not confined to a single metric but are equally valid within the realm of semantic-level evaluation. Moreover, we observed that in the audio modality setting, BLEURT scores were relatively lower. This is likely because BLEURT is more sensitive to surface-level textual variations, mismatches in phrasing, and text length; even when the underlying semantic content is entirely accurate, such variability can still lead to a decline in BLEURT scores.
> > >
> > > **The Link :** https://anonymous.4open.science/r/icml26_rebuttal_537-24FD/Rebuttal__for_Reviewer__bXDU.pdf

---

### Decision · Program_Chairs · 2026-04-30

**Decision:**

Accept (regular)

**Comment:**

The paper addresses an import problem in MFL, of which the technology is novel and sound. Extensive experiments also demonstrate its effectiveness.